# Acetylation discriminates disease-specific tau deposition

Pijush Chakraborty [1], Gwladys Rivière[1], Alina Hebestreit [2], Alain Ibáñez de Opakua [1], Ina M. Vorberg [2,3], Loren B. Andreas[4] & Markus Zweckstetter [1,4] ✉

Pathogenic aggregation of the protein tau is a hallmark of Alzheimer's disease and several other tauopathies. Tauopathies are characterized by the deposition of specific tau isoforms as disease-related tau filament structures. The molecular processes that determine isoform-specific deposition of tau are however enigmatic. Here we show that acetylation of tau discriminates its isoform-specific aggregation. We reveal that acetylation strongly attenuates aggregation of four-repeat tau protein, but promotes amyloid formation of three-repeat tau. We further identify acetylation of lysine 298 as a hot spot for isoform-specific tau aggregation. Solid-state NMR spectroscopy demonstrates that amyloid fibrils formed by unmodified and acetylated three-repeat tau differ in structure indicating that site-specific acetylation modulates tau structure. The results implicate acetylation as a critical regulator that guides the selective aggregation of three-repeat tau and the development of tau isoform-specific neurodegenerative diseases.

Alzheimer's Disease and other neurodegenerative diseases are age-related disorders characterized by synaptic dysfunction and the progressive loss of neurons[1]. The microtubule-associated protein tau is the major constituent of insoluble deposits found in the brain of patients affected by neurodegenerative diseases named tauopathies. Although six different isoforms are present in the adult human brain[2], different tau isoforms aggregate into insoluble deposits in distinct tauopathies[3]. In addition, tau fibrils extracted from brains of Alzheimer's Disease patients have a different structure than those extracted from patient brains affected by the tauopathies Corticobasal Degeneration (CBD) and Pick's Disease[3]. Clinical phenotypes and disease courses may thus be connected to tau isoform and amyloid structure[3]. In addition, post-translational modifications and so-far unknown co-factors may critically influence tau fibril structure[4]. How tau isoforms, tau aggregate structure and post-translational modifications are related and how they determine disease- and isoform-specific tau deposition in tauopathies is however unknown.

Human tau is encoded by the microtubule-associated protein tau gene located on chromosome 17q21[5]. Six isoforms of tau are generated due to alternative splicing of exons 2, 3, and 10. Exons 2 and 3 each encode an insert of 29 amino acids at the amino terminus generating isoforms containing 0, 1, or 2 inserts. In addition, Tau contains four imperfect, but highly conserved repeats of 30–31 amino acids, encoded by exons 9–12. Alternative splicing of exon 10, which encodes the second pseudo-repeat (R2), generates isoforms that contain either three (R1, R3, and R4) or four repeats (R1, R2, R3, and R4) and are named 3R tau and 4R tau, respectively. In the brain of healthy human adults, the expression of 3R and 4R tau is comparable. Mutations in the tau gene can change the abundance of 3R and 4R tau[6]. 3R and 4R tau isoforms differ in their ability to regulate microtubule dynamics[7]. Insoluble deposits of tau can consist of both 3R and 4R tau isoforms, such as in Alzheimer's disease, or only one of the two isoforms[1,3]. In Pick's Disease, only 3R tau is found in aggregates, hence the name 3R tauopathy[1,8].

Tau undergoes extensive post-translational modifications (PTMs) in the brain with up to 35% of residues being susceptible to modification[9]. In support of an important role of acetylation, histone acetyltransferases are dysregulated in tauopathies[10,11]. For example,

[1]German Center for Neurodegenerative Diseases (DZNE), Von-Siebold-Str. 3a, 37075 Göttingen, Germany. [2]German Center for Neurodegenerative Diseases (DZNE), Bonn, Germany. [3]Rheinische Friedrich-Wilhelms-Universität, Bonn, Germany. [4]Department for NMR-based Structural Biology, Max Planck Institute for Multidisciplinary Sciences, Am Faßberg 11, 37077 Göttingen, Germany. ✉e-mail: markus.zweckstetter@dzne.de

increased activity of p300/CBP was identified in FTLD-tau patients' brains, whereas both the activity and levels of p300/CBP are reduced in the brain of Alzheimer's disease patients[11,12]. In addition, distinct acetylation patterns have been detected in different tauopathies[13,14]. Acetylated tau also inhibits tau clearance by chaperone-mediated autophagy, promoting tau pathology[15,16]. In agreement with a disease-associated role of tau acetylation, decreasing the amount of acetylated tau by inhibiting the acetyltransferase activity of p300/CBP is neuroprotective in brain injury[17].

Here we provide insights into the molecular determinants of isoform-specific tau and the enigmatic emergence of 3R tauopathies. Using a combination of biochemical experiments, aggregation assays, site-directed mutagenesis, in vitro enzymatic reactions and solid-state NMR spectroscopy, we define a specific role of acetylation in driving 3R tau specific aggregation and amyloid structure.

## Results

### 3R tau aggregates into amyloid fibrils without co-factors

In Pick's disease, only the 3R tau isoforms aggregate into insoluble deposits[18]. To gain insight into 3R-specific aggregation, we recombinantly produced the 352-residue 0N3R tau protein (Fig. 1a, b; hereafter named 3R tau), and subjected it to in vitro aggregation. Aggregation was performed at 37 °C in the absence of heparin or other co-factors using an assay previously established for 4R tau[19]. 3R tau started to form fibrils after two days of incubation according to the increase in thioflavin-T (ThT) fluorescence intensity (Fig. 1c). At the end of the incubation period, the samples displayed circular dichroism (CD) spectra typical for amyloid fibrils with a minimum at ~218 nm (Supplementary Fig. 1a). Negative-stain electron microscopy (EM) confirmed the presence of fibrils (Fig. 1d).

To determine the rigid core of the 3R tau fibrils, we aggregated uniformly $^{13}C/^{15}N$-labeled protein and recorded $^{1}H$-$^{15}N$ J-transfer spectra with magic angle spinning (Fig. 1e). This experiment uses J-coupling to transfer the magnetization and selectively detect highly dynamic residues. Sequence-specific analysis identified residues upstream of residue ~260 and downstream of residue ~380 (Fig. 1e, f). Cross-peaks for residues in between were not observed (Fig. 1f), indicating that the rigid core of the 3R tau fibrils comprises approximately residues 260–380. In addition to residues in the central part of tau, ~30–40 residues at the N-terminus were broadened beyond detection in the fibril spectrum. This stretch of residues contains the epitope for recognition of pathologically aggregated tau by conformation-specific monoclonal antibodies[20]. In the cryoEM structure of tau fibrils extracted from the brain of patients with Pick's disease, residues 254–378 were resolved[8].

To confirm the NMR analysis, 3R tau fibrils were also digested by trypsin to remove the fuzzy coat, followed by pelleting down the trypsin-resistant material and loading it in an SDS-PAGE gel. We analyzed the trypsin-resistant band observed in the SDS-PAGE gel by mass spectrometry and detected peptides from residues ~250 to 438 (Supplementary Fig. 1b). The detection of peptides for the C-terminal residues (after residue 380) maybe due to the scarcity of residues (K395, R406, K438) targeted by trypsin. We therefore repeated the digestion experiment in the presence of the more promiscuous protease pronase. We detected peptides up to residue 380 when analyzing the pronase-resistant band by mass spectrometry (Fig. 1g). The combined data show that 3R tau efficiently aggregates in vitro into amyloid fibrils that have similar residues in the rigid core as fibrils extracted from Pick's disease patient brain.

### Acetylation of tau at repeats R2/R3

Brain-derived 3R tau fibrils are acetylated at several of the 37 lysine residues of 3R tau (Fig. 2a)[14]. To determine the sites of acetylation of 3R tau with single-residue resolution and quantify their degree of acetylation with high accuracy, we performed in vitro acetylation of

$^{15}N$-lysine labeled, monomeric 3R tau in the presence of either p300 or CBP, or both enzymes combined. The acetylation levels of individual lysine residues were determined from the intensity ratios of the cross-peaks of the unacetylated lysine (in the acetylated sample) and unmodified lysine in 2D $^{1}H$-$^{15}N$ HSQC NMR spectra (Fig. 2b). The analysis revealed broad acetylation of the lysine residues in 3R tau (Supplementary Fig. 2). The degree of acetylation strongly varied from ~20 to 100% between different lysine residues (Supplementary Fig. 2). The acetylation levels of lysine residues that are very close in the protein sequence (K224/K225, K280/K281, K369/K370) could not be determined precisely due to the splitting of resonances into multiple peaks. Tau acetylation was similar when p300 and CBP were used individually or in combination (Supplementary Fig. 2a, c).

To test if acetylation is isoform-specific, we acetylated 4R tau (Fig. 2a). Mass spectrometric analysis of the acetylated 4R tau revealed that almost all lysine residues are acetylated (Supplementary Fig. 3a, Supplementary Data 1). Residue-specific analysis showed that the levels of acetylation of individual lysine residues are similar in 3R and 4R tau with K317 and K321 being most strongly acetylated (Supplementary Fig. 2a, c). In addition, the five lysine residues (K280, K281, K290, K294, K298) of repeat R2—the repeat that is missing in 3R tau isoforms—were strongly acetylated in 4R tau (Supplementary Figs. 2a, c and 3b).

The two cysteine residues present in the repeats R2 and R3 of tau are reported to have catalytic activity promoting auto-acetylation of tau in the absence of any acetyltransferases[21,22]. To determine the site-specific auto-acetylation levels of 3R and 4R tau we characterized the in vitro acetylation of $^{15}N$-lysine labeled, monomeric 3R/4R tau in the absence of acetyltransferases, i.e., only in the presence of acetyl-coA. Under the condition of auto-acetylation, the lysine residues nearby to the two cysteine residues (C291, C322) of tau reached a comparable level of acetylation as in the presence of acetyltransferases (Supplementary Fig. 2a, c). This suggests that the strong acetylation of the five lysine residues of repeat R2 was mostly due to the auto-acetylation of tau. However, the lysine residues away from the cysteine residues of tau were ~10–20% more acetylated in the presence of acetyltransferases (Supplementary Fig. 2b, d).

To identify the most reactive lysine residues of tau, we acetylated 4R tau in the presence of both P300 and CBP acetyltransferases for 2 h. The reduction of reaction time from 12 to 2 h leads to a global decrease in acetylation levels (Supplementary Fig. 4a). However, the lysine residues nearby to the two cysteine residues of tau (C291, C322) were efficiently acetylated.

Physiological tau is bound to microtubules in axons[23]. We therefore repeated the acetylation reactions (both enzymatic and auto-acetylation) of $^{15}N$-lysine-labeled 4R tau in the presence of a two-fold excess of microtubules over tau, matching the molar stoichiometry of the tau:tubulin interaction[24]. In addition, we decreased the time of acetylation from 12 to 2 h, in order to acetylate predominantly the most reactive lysines in tau. Under these conditions, only three lysine residues in repeat R2 (K290, K294, K298) and two lysine residues in repeat R3 (K317, K321) were efficiently acetylated (>40%) (Fig. 2d). In contrast, K280 and K281, which are part of the heptapeptide motif in R2, which localizes to the intra-dimer interface of microtubules[25], only reached low levels of acetylation (Fig. 2d). The combined data reveal selective acetylation of five lysine residues in tau, with three of them (K290, K294 and K298 in repeat R2) unique to 4R tau.

### Acetylation accelerates 3R tau but strongly attenuates 4R tau fibrillization

The five lysine residues that are efficiently acetylated are part of repeat R2 and R3, which play an important role in the pathogenic aggregation of tau[26–28]. Tau acetylation thus might have a strong impact on both the protein's aggregation kinetics and the amyloid fibril structure. To gain insight into the influence of acetylation on tau fibrillization, we

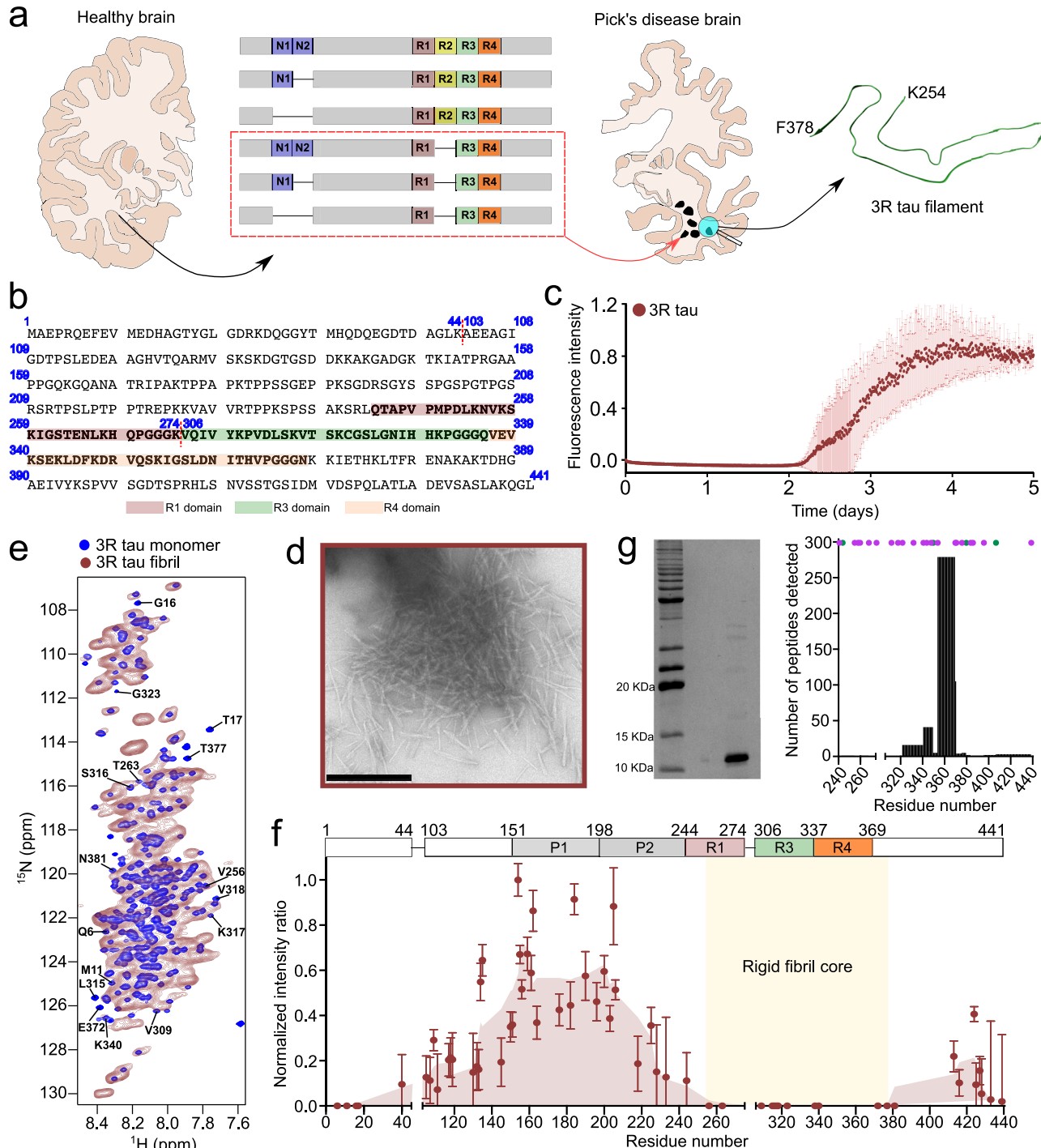

**Fig. 1 | Co-factor-free aggregation of 3R tau. a** Schematic representation of the specific deposition of 3R tau in Pick's disease. In the healthy brain, all six isoforms of tau are present whereas in Pick's disease brain only 3R isoforms of tau (red box) are deposited. The structure of the tau filament extracted from the brain of a Pick's disease patient (PDB code: 6GX5) is shown. **b** Amino acid sequence of 0N3R tau. The amino acids are numbered based on the sequence of 2N4R tau. The pseudo-repeat domains R1, R3, and R4 are highlighted in light red, green, and orange, respectively. **c** Fibrillization kinetics of 3R tau (25 μM) in the absence of co-factors. Error bars represent std of $n = 3$ independent samples. The center of the error bars represents the average value of $n = 3$ independent samples. Source data are provided as a Source data file. **d** Negative-stain electron micrographs of 3R tau fibrils. Scale bar, 500 nm. The micrograph is representative of $n = 3$ biological replicates. **e, f** NMR analysis of the rigid core of 3R tau fibrils. Superposition of the ¹H-¹⁵N HSQC spectrum of monomeric 3R tau (blue, **e**) with ¹H-¹⁵N J-transfer MAS spectra of 3R tau fibrils (red, **e**). Assignments of residues broadened beyond detection in the fibrils are displayed. Residue-specific intensity ratios derived from (**e**) are shown in (**f**). Errors in intensity ratios were calculated from the signal-to-noise ratio of the cross-peaks in the respective spectra. Smoothed data are shown in light brown. Residues resolved in the cryoEM structure of tau fibrils extracted from the brain of a patient with Pick's disease (PDB code: 6GX5) are indicated by yellow shading. Source data are provided as a Source data file. **g** Protease digestion of 3R tau fibrils. SDS-PAGE gel of pronase-digested 3R tau fibrils. The protease digestion experiment has been performed up to 3 times with similar result. Numbers of detected peptides are shown to the right. The position of lysine and arginine residues are marked with purple and green dots, respectively. Source data are provided as a Source data file.

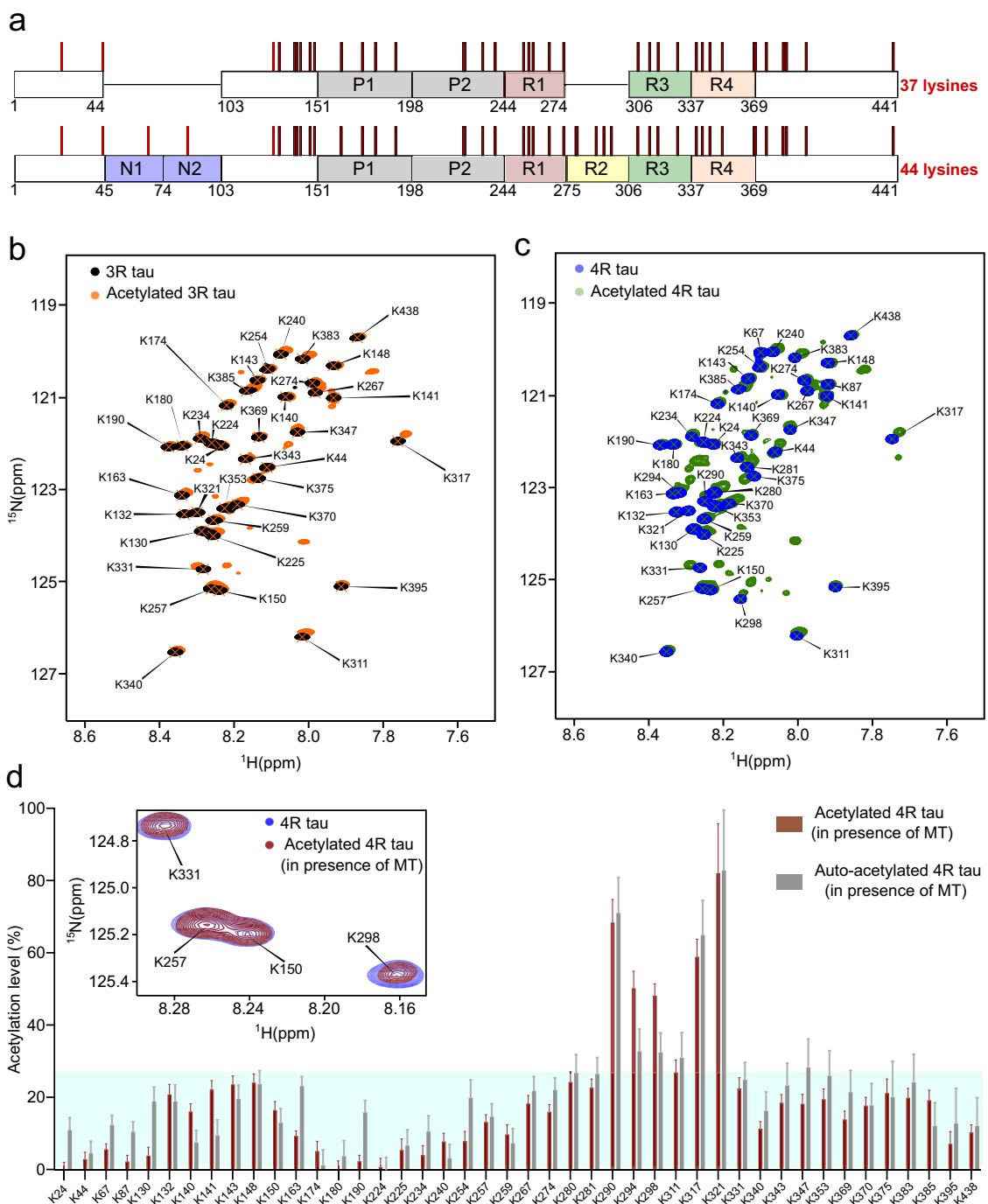

**Fig. 2 | Single-residue analysis of tau acetylation. a** Domain diagram of 3R and 4R tau. Lysines are indicated by red bars. The total numbers of lysine residues present in 0N3R tau and 2N4R tau are indicated. **b**, **c** Superposition of the ¹H-¹⁵N HSQC spectra of unmodified lysine-labeled 3R tau (**b**; 50 μM; black) or 4R tau (**c**; 50 μM; blue) and the corresponding acetylated proteins (**b**: orange; **c**: green). Acetylation was performed by incubation with both p300 and CBP for 12 h. **d** Analysis of acetylation levels of individual lysine residues of 4R tau after acetylating with either both p300 and CBP (brown), or in the absence of any acetyltransferases (auto-acetylation) (gray) for 2 h in the presence of microtubules. Errors in acetylation levels were calculated from the signal-to-noise ratio of the cross-peaks in the NMR spectra. The cyan box represents a 25% cut-off for weakly acetylated lysine residues. The inset displays a superposition of a selected region of the ¹H-¹⁵N HSQC spectra of unmodified lysine-labeled 4R tau (blue) and acetylated 4R tau (in presence of microtubules) (brown). Source data are provided as a Source data file.

performed aggregation assays of unmodified and acetylated 3R tau. Unmodified 3R tau started to fibrillize after ~2–2.5 days (Fig. 3a). In contrast, acetylated 3R tau forms fibrils already after ~1.5 days (Fig. 3a), demonstrating that acetylation strongly accelerates the fibrillization of 3R tau.

We then also incubated unmodified and acetylated 4R tau in aggregation-promoting conditions. Unmodified 4R tau displayed increasing ThT fluorescence, similar to unmodified 3R tau (Fig. 3a).

However, the situation drastically changed when tau was acetylated: acetylated 4R tau did not display an increase in ThT fluorescence when incubated for even five days (Fig. 3a, b).

To confirm the findings from the aggregation assay, we recorded CD spectra of the different proteins after five days of aggregation. Both the unmodified and acetylated 3R tau displayed a typical spectrum for amyloid fibrils with a minimum at ~218 nm (Fig. 3c). Electron microscopy confirmed the presence of fibrils (Fig. 3e). We also observed a

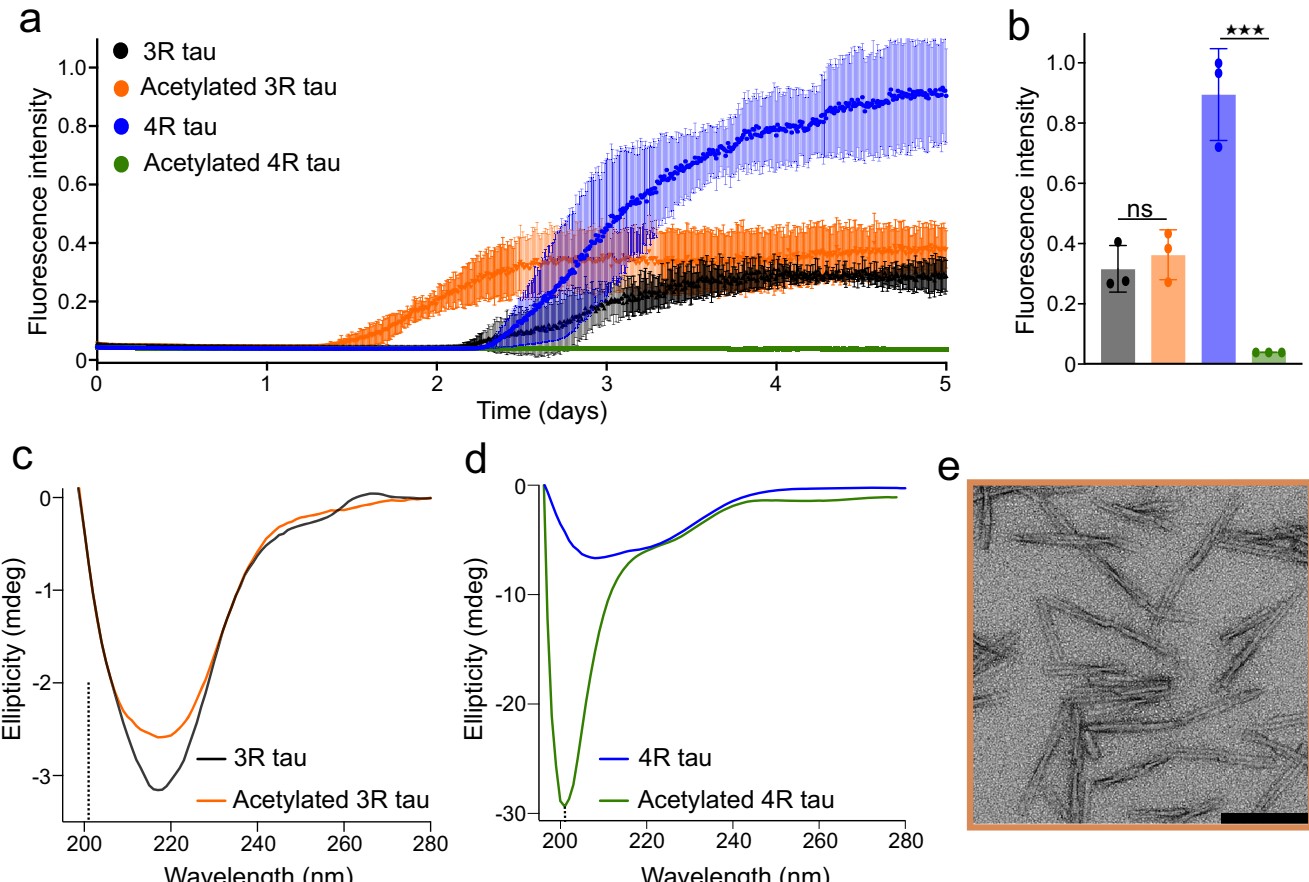

**Fig. 3 | Acetylation accelerates 3R tau but strongly attenuates 4R tau fibrillization. a** Fibrillization kinetics of unmodified 3R (black) and 4R (blue) tau, as well as acetylated 3R (orange) and 4R (green) tau followed by ThT fluorescence. Acetylation reactions were performed in the presence of p300/CBP. Protein concentrations were 25 μM. Error bars represent std of $n = 3$ independent samples. The center of the error bars represents the average value of $n = 3$ independent samples. The ThT fluorescence experiment of both acetylated 3R and 4R tau has been performed up to 5 times with 3 different batches of tau. Before performing each of the ThT fluorescence experiments, the acetylation reaction has been performed independently. In all cases the data were reproducible. Source data are provided as a Source data file. **b** Final ThT intensity of unmodified 3R (black) and 4R (blue) tau, as well as acetylated 3R (orange) and 4R (green) tau after five days of aggregation.

Acetylation reactions were performed in the presence of p300/CBP. Statistical analyses were performed by two-tailed Welch's t test (***$p = 0.006$). Error bars represent std of $n = 3$ independent samples. The center of the error bars represents the average value of $n = 3$ independent samples. Source data are provided as a Source data file. **c** CD spectra of unmodified (black) and acetylated (orange) 3R tau after five days of aggregation. The location of the minimum expected for random coil structure is marked by a dotted line. Source data are provided as a Source data file. **d** CD spectra of unmodified (blue) and acetylated (green) 4R tau after five days of aggregation. Source data are provided as a Source data file. **e** Negative-stain EM of acetylated 3R tau fibrils. The micrograph is representative of $n = 3$ biological replicates. Scale bar, 200 nm.

typical β-structure CD spectrum for the aggregated unmodified 4R tau (Fig. 3d). In contrast, a random coil CD spectrum with a minimum at ~200 nm was seen for the acetylated 4R protein, confirming that acetylated 4R tau does not form amyloid fibrils even after long periods of incubation (Fig. 3d).

To study how the duration of the acetylation reaction, as well as the acetylation levels of individual lysine residues, impacts the aggregation of 4R tau, we aggregated the 4R tau acetylated in the presence of P300 and CBP acetyltransferases for 2 h. The 2-h acetylation also strongly delayed the aggregation of 4R tau (Supplementary Fig. 4b, c) and lead to the aggregation of only ~20% acetylated protein (Supplementary Fig. 4d, e). This is in agreement with our findings that prolonged acetylation (12 h) of 4R tau inhibits its aggregation (Fig. 3a).

### K298 acetylation drives isoform-specific accumulation of tau
The above data reveal that acetylation strongly accelerates fibril formation of 3R tau while at the same time strongly attenuating 4R tau aggregation (Fig. 3). We can rationalize the faster aggregation of 3R tau upon acetylation when considering the importance of electrostatic interactions in tau aggregation: acetylation of lysine residues removes

positive charges which decreases electrostatic repulsion and favors intermolecular interactions of the repeat region during tau aggregation. However, simple changes in electrostatics would also predict faster aggregation of acetylated 4R tau, in striking contrast to the observed inhibition of 4R tau aggregation by acetylation (Fig. 3a). Notably, the only detectable difference in acetylation between 3R and 4R tau is the acetylation of the five lysine residues in repeat R2 of 4R tau (Fig. 2, Supplementary Fig. 2c). We thus hypothesize that the acetylation of lysine residues in R2, the repeat that is unique to 4R tau, is a critical determinant in isoform-specific tau accumulation.

To study the role of acetylation of individual lysine residues in discriminating between 3R and 4R tau aggregation, we created several lysine-to-glutamine mutants of 4R tau. We individually mutated all five lysine residues (K280Q, K281Q, K290Q, K294Q, K298Q) of repeat R2 (Fig. 4a). In addition, we created the double mutant K298Q/K311Q because previous studies suggested an important pathogenic role of acetylation of K311[28]. We then aggregated the six mutant proteins along with wild-type 4R tau (Supplementary Fig. 5a).

Analysis of the aggregation kinetics showed that five out of the six K-to-Q mutant proteins aggregated faster than wild-type 4R tau

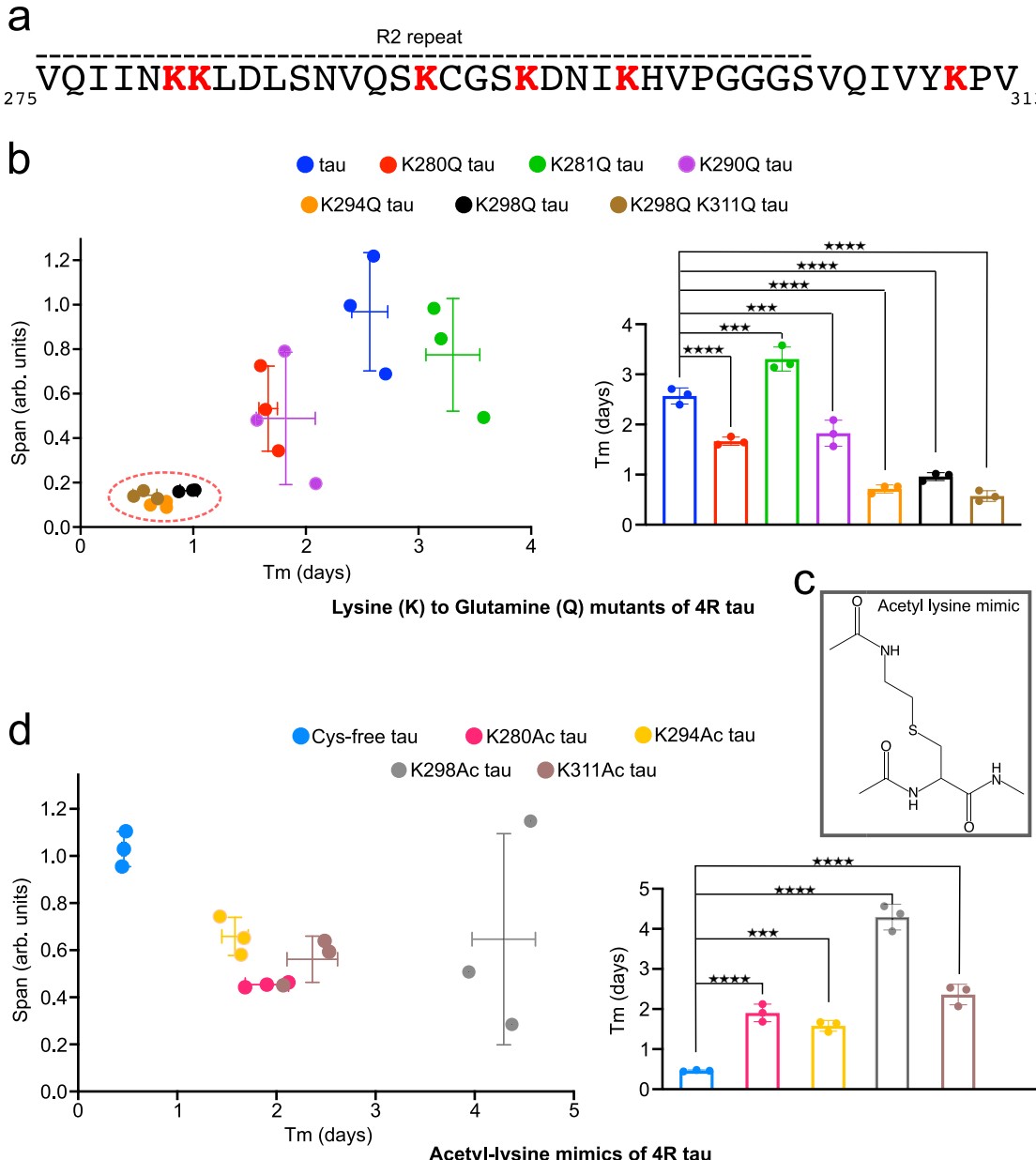

**Fig. 4 | K298 acetylation delays 4R tau fibrillization. a** Amino acid sequence of the R2 repeat and the beginning of repeat R3 of tau. Lysine residues are shown in red. **b** Impact of lysine-to-glutamine mutation in repeat R2/R3 on 4R tau fibrilliza-tion. (left) ThT intensity span vs. half-time of aggregation (Tm). Error bars represent std of $n = 3$ independently aggregated samples. The center of the error bars represents the average value of $n = 3$ independent samples. (right) Half-time of aggregation (Tm) of different lysine-to-glutamine tau mutants; statistical analysis by one-way ANOVA: ****$p < 0.0001$/***$p = 0.003$. Error bars represent std of $n = 3$

independent samples. The center of the error bars represents the average value of $n = 3$ independent samples. Source data are provided as a Source data file. **c, d** Aggregation kinetics of cysteine-free 4R tau acetylated at single lysine-mimic residues. This lysine acetylation mimic differs only by the presence of a sulfur atom in place of the Cγ of lysine (**c**). Error bars represent std of $n = 3$ independently aggregated samples. The center of the error bars represents the average value of $n = 3$ independent samples. Statistical analysis by one-way ANOVA: ****$p < 0.0001$/ ***$p = 0.003$. Source data are provided as a Source data file.

(Fig. 4b). Only in the case of K281Q tau, the half-time of fibrillization was increased while maintaining a similar maximum ThT intensity (Fig. 4b). The enhanced aggregation kinetics, which we observed for the five K-to-Q tau mutants (K280Q, K290Q, K294Q, K298Q, K298Q/ K311Q), can again be rationalized on the basis of a change in electro-statics: the mutation-associated decrease in positive charge in the repeat region lowers the electrostatic repulsion between R2 repeats during the aggregation of 4R tau.

In addition to the changes in aggregation kinetics, we observed a strong decrease in the maximum ThT intensity reached by the three tau mutants K294Q, K298Q, and K298Q/K311Q (Fig. 4b, Supplemen-tary Fig. 5a). To evaluate if the lower ThT intensity is due to attenuated

aggregation, we centrifuged all samples at the end of the incubation period. The supernatants were loaded into an SDS-PAGE gel and the amount of residual soluble protein was analyzed (Supplementary Fig. 6a). Comparison of the intensity of the supernatant to the monomeric unaggregated protein, showed that comparable protein amounts were aggregated for the wild-type 4R tau and the K-to-Q mutant proteins (Supplementary Fig. 6a). This suggests that the lower ThT intensities detected for K294Q, K298Q and K298Q/K311Q mutant tau are not a result of aggregation inhibition, but might arise from differences in tau fibril structure induced by the mutations.

K-to-Q mutation is widely used to mimic acetylation of proteins in cell and animal studies[29,30]. The mutation captures the acetylation-

associated removal of the positive charge of lysine, but fails to represent the size of the acetylated lysine side chain (Supplementary Fig. 7). We therefore created four single-site acetylation mimics (K280Ac, K294Ac, K298Ac, K311Ac) of 4R tau by mutating individual lysine residues in R2 to cysteine, followed by converting the cysteine residue to dehydroalanine, which enables the access of N-acetylcysteamine (Supplementary Fig. 7)[31]. K281 and K290 were not included, because they did not strongly affect aggregation in the analysis of the K-to-Q mutations (Fig. 4b). Prior to introduction of the lysine-to-cysteine mutations, the two endogenous cysteine residues (C291, C322) were changed to serine. The acetylation mimics created using this protocol are similar to acetyl-lysine and differ only by the presence of a sulfur atom in place of the Cγ of lysine (Fig. 4c). The formation of the acetylation mimics was confirmed by mass spectrometry (Supplementary Fig. 8).

We then aggregated the 4R tau proteins containing the acetylation mimics, as well as the cysteine-free 4R tau as reference (Supplementary Fig. 5b). In agreement with previous studies, we observed that cysteine removal strongly accelerates 4R tau aggregation (cysteine-free tau in Supplementary Fig. 5b when compared to wild-type tau in Supplementary Fig. 5a), because intermolecular disulfide bonds no longer interfere with aggregation[32]. Detailed analysis of the aggregation kinetics showed that all four acetylation mimics strongly delay the fibrillization of 4R tau (Fig. 4d, Supplementary Fig. 5b). The delay in fibril formation was strongest for the K298Ac mimic, increasing the duration of the fibrillization lag phase by ~8-fold (Fig. 4d). The other three acetylation mimics (K280Ac, K294Ac, K311Ac) delayed the onset of fibrillization by a factor of ~3–5 (Fig. 4d), i.e., were intermediate between the cysteine-free tau and K298Ac tau. In addition, all four acetylation mimics had lower maximum ThT intensity and less aggregated protein at the end of the aggregation period (Fig. 4d, Supplementary Fig. 6b). The combined data show that the lysine residues in repeat R2 critically influence aggregation of 4R Tau. While acetylation of a single lysine in repeat R2 is not sufficient to fully block aggregation, acetylation of K298 most strongly delays aggregation of 4R tau.

### Structural characterization of the acetylated 3R tau fibrils

Our findings that acetylation promotes aggregation of 3R tau, but strongly attenuates aggregation of 4R tau, suggest that tau proteins, which aggregate in the 3R tauopathy Pick's disease, are acetylated. To characterize the amyloid fibrils of acetylated 3R tau, we performed pronase digestion, followed by pelleting down the pronase-resistant core (Fig. 5a). Mass spectrometry of the pronase-resistant band revealed that the acetylated 3R tau fibrils have similar residues in the rigid core as unmodified 3R tau fibrils, and as tau fibrils purified from the brain of patients with Pick's disease (Figs. 5a, 1g)[8].

To further investigate the structural properties of acetylated 3R tau fibrils, we prepared $^{13}C/^{15}N$-labeled acetylated 3R tau and performed proton-detected solid-state NMR spectroscopy (Fig. 5b, c). For comparison, we also recorded solid-state NMR spectra of unmodified 3R tau (Fig. 5b, c). Figure 5 shows a superposition of the proton-detected 2D hNH spectra of acetylated and unmodified 3R tau, as well as the overlap of the 2D $^{15}N$-$^{13}C$ planes from 3D (H)CANH experiments. While we observed well-resolved cross-peaks for both fibrils, the signals of the cross-peaks of the acetylated 3R tau fibrils were generally broader than those of the unmodified fibril, potentially due to heterogenous acetylation. Closer inspection of the 2D $^{15}N$-$^{13}C$ planes further showed that some of the cross-peaks of the acetylated 3R tau fibrils have the same chemical shift as the unmodified 3R tau fibrils (Fig. 5c). At the same time, several cross-peaks were shifted in the spectrum of the acetylated 3R tau fibrils (Fig. 5c). In addition, new cross-peaks appeared in the spectra of the acetylated 3R tau fibrils. The solid-state NMR spectra therefore suggest that some structural similarity exists between the acetylated and unmodified 3R tau fibrils, but

plausibly their structure is not the same. However, as we lack the residue-specific resonance assignments, variation of the NMR signals in the acetylated state might also be simply due to the effect of acetylation on the resonance of lysine and neighboring residues.

## Discussion

Acetylation has emerged as an important post-translational modification of tau in Alzheimer's disease[26,27] and other tauopathies[26,33,34]. Acetylation of K280 is detected in the diseased brain but not in the brain of healthy individuals[26,27,35]. It decreases the binding affinity of tau to microtubules, which might enhance the aggregation of tau into paired helical filaments[26]. Acetylation of K274 and K281 has been identified in AD brains and was reported to disrupt synaptic plasticity by reducing the Kidney/Brain protein[36]. Acetylation of K274 and K281 also destabilizes the axon initial segment and promotes mislocalization of tau into the somatodendritic compartment initiating an early event of neurodegeneration[37]. In addition, acetylation of K174 may be an early modification in Alzheimer's disease patients[10], and the expression of pseudo-acetylated tau at K174 attenuated tau clearance in transgenic mice[10]. Consistent with a critical role of acetylation of tau in the pathogenic progression of different tauopathies, we here show that acetylation discriminates isoform-specific deposition of tau.

To study the effect of acetylation on the aggregation of the 3R isoform of tau, it is essential to aggregate the protein in the absence of co-factors, which otherwise may override the impact of post-translational modifications. We therefore subjected 3R tau to a previously established co-factor-free aggregation assay, which enabled aggregation of full-length 4R tau[19], and showed that the assay efficiently converts 3R tau into amyloid fibrils (Fig. 1). Solid-state NMR spectroscopy and protease digestion demonstrated that the location of the fibrillar core of the in vitro generated co-factor-free 3R tau fibrils is similar to the core of tau filaments derived from the brain of patients with the 3R tauopathy Pick's disease (Fig. 1).

We then used the co-factor free aggregation assay, to study the impact of acetylation on tau amyloid formation. We found that acetylation promotes fibril formation of 3R tau, but strongly attenuates fibrillization of 4R tau (Fig. 3). Because 3R and 4R tau only differ with respect to repeat R2, the data suggest that acetylation in repeat R2 plays a critical role in inhibiting the aggregation of 4R tau. The lysine residues that were most strongly acetylated in repeat R2 were K290, K294, and K298 (Fig. 2d). Employing single-site acetylation mimics, we further found that the simultaneous acetylation of multiple lysine residues in R2 is needed to block aggregation of 4R tau (Fig. 4).

Using site-specific pseudo-acetylation, we showed that acetylation of K298 most strongly delays aggregation of 4R tau (Fig. 4d). Inspection of the 3D structure of tau filaments purified from the 4R tauopathy CBD provides a rationale for the importance of K298 acetylation in discriminating 4R and 3R tauopathies (Fig. 6c). In the tau CBD structure[13,38], the side chain of K298 points inside the core of the tau fibrils and forms a salt bridge with D358. When K298 is acetylated the salt bridge can no longer form. In addition, the increased bulkiness of the side chain of K298 upon acetylation will generate steric clash, i.e., 4R tau will be unable to form the CBD structure when K298 is acetylated. The impact of acetylation at K294 on 4R tau aggregation can further be rationalized on the basis of the structure of tau fibrils purified from the 4R tauopathy Progressive Supranuclear Palsy (PSP; Fig. 6d): K294 forms a salt bridge with D314/S316 in the core of the tau PSP structure[3,39]. Thus, the delay observed in the aggregation of 4R tau upon specifically acetylating K298 or K294 likely arises from the increase in steric crowding in the vicinity of these residues upon acetylation. Thus, acetylation of specific lysine residues in repeat R2 favors the accumulation of 3R tau.

Besides acetylation in repeat R2, acetylation of other lysine residues may contribute to isoform-specific tau deposition[14]. For example, K311 is acetylated in the 3R tauopathy Pick's disease, whereas it is not

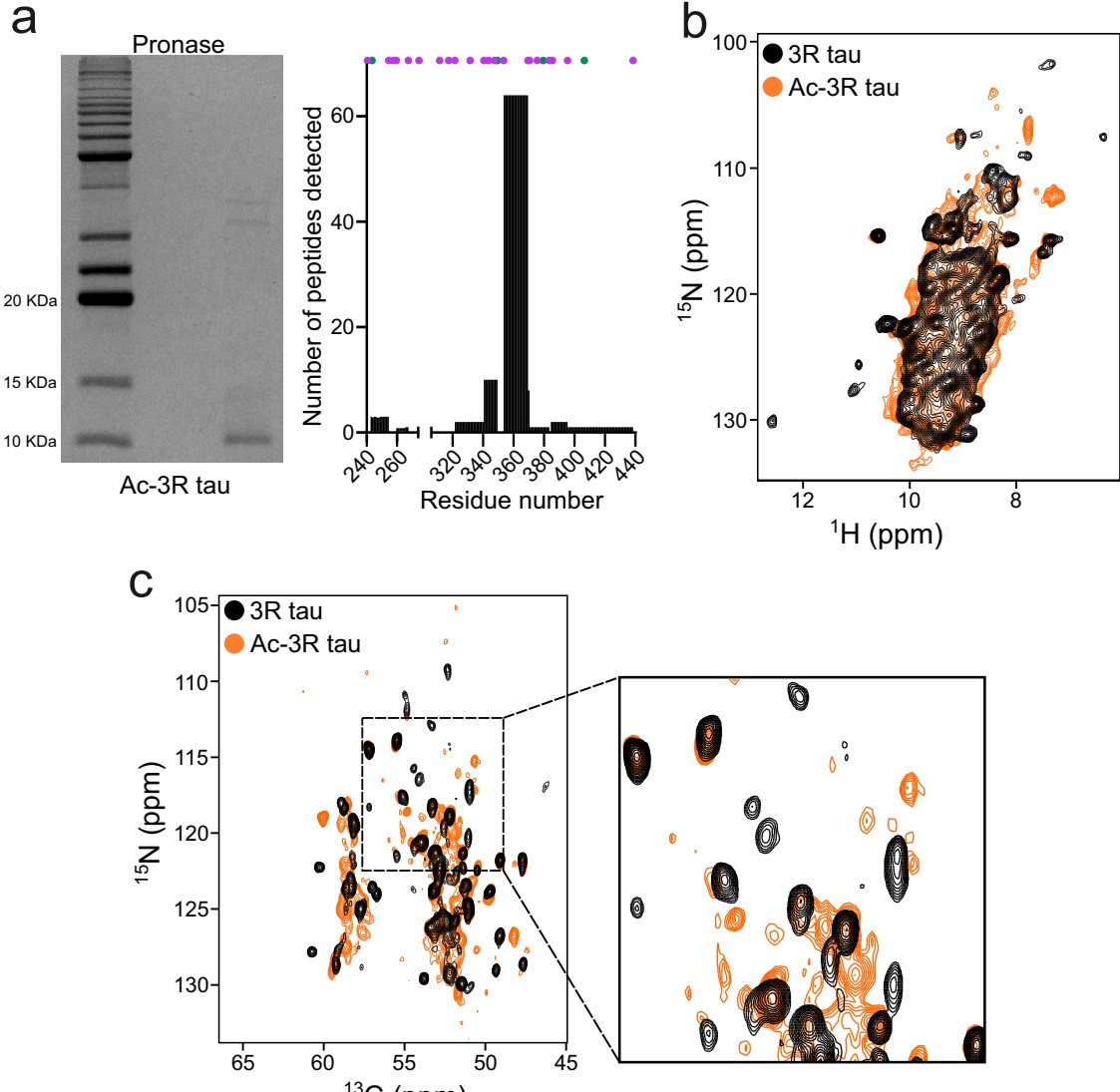

**Fig. 5 | Structural characterization of acetylated 3R tau fibrils. a** SDS-PAGE gel of pronase-digested acetylated 3R tau fibrils (left). The protease digestion experiment has been performed upto 3 times with similar result. Number of peptides detected from the enzymatic digestion of the tau band (right). Lysine and arginine residues are marked with purple and green dots, respectively. Source data are provided as a Source data file. **b, c** Superposition of hNH spectra (**b**) and 2D $^{15}$N-$^{13}$C planes of 3D (H)CANH spectra (**c**) of unmodified (black) and acetylated (orange) 3R tau fibrils.

acetylated in the case of the 4R tauopathies CBD and PSP[14,28]. In the structure of tau fibrils from Pick's disease[8], K311 points away from the filament core (Fig. 6b). Acetylation of K311 thus will not interfere with the structure of the tau fibril core in Pick's disease. In contrast, K311 forms a salt bridge with D295 in the CBD tau structure (Fig. 6c), and is located in a compact region of the PSP tau structure (Fig. 6d). Selective acetylation of K311 in Pick's disease is thus likely, because acetylation of K311 may interfere with the formation of CBD/PSP tau fibril structures due to steric hindrance. In agreement with this hypothesis, we observed strong inhibition of 4R tau aggregation upon selectively acetylating K311 (Fig. 4d). In addition, selective acetylation of K311 in Pick's disease will be favored by its solvent accessible location in the structure of the Pick's disease fibrils, whereas the K311 side chain is not accessible in the structure of tau fibrils from CBD and PSP (Fig. 6c, d).

The combined data establish a model that rationalizes the selective deposition of 3R tau in 3R tauopathies (Fig. 6e). In the adult human brain, both 3R and 4R isoforms of tau are present. However, during the disease both isoforms become acetylated. This accelerates pathogenic accumulation of 3R tau, and at the same time

blocks aggregation of 4R tau. Thus, insoluble deposits of only 3R tau build up in the brain of patients with Pick's disease. As we were unable to determine the structure of the in vitro acetylated 3R tau fibrils, the interpretations are only rationalizations and models, to be tested by future experiments.

Taken together, our results suggest that tau acetylation is a key determinant in the emergence of 3R tauopathies. The findings also lay the foundations to investigate the molecular factors that determine the formation of distinct tau strains in 4R tauopathies, as well as in Alzheimer's disease and other tauopathies where both 3R and 4R tau accumulate into insoluble deposits.

## Methods
### Protein preparation
Wild-type as well as all mutants of tau were cloned into pNG2 vector (a derivative of pET-3a, Merck-Novagen, Darmstadt) within Bam H1 and Nde1 restriction sites. The pNG2-2N4R tau plasmid was kindly provided by Eckhard Mandelkow[40]. Recombinant proteins were expressed in *Escherichia coli* strain BL21(DE3) supplied by Novagen.

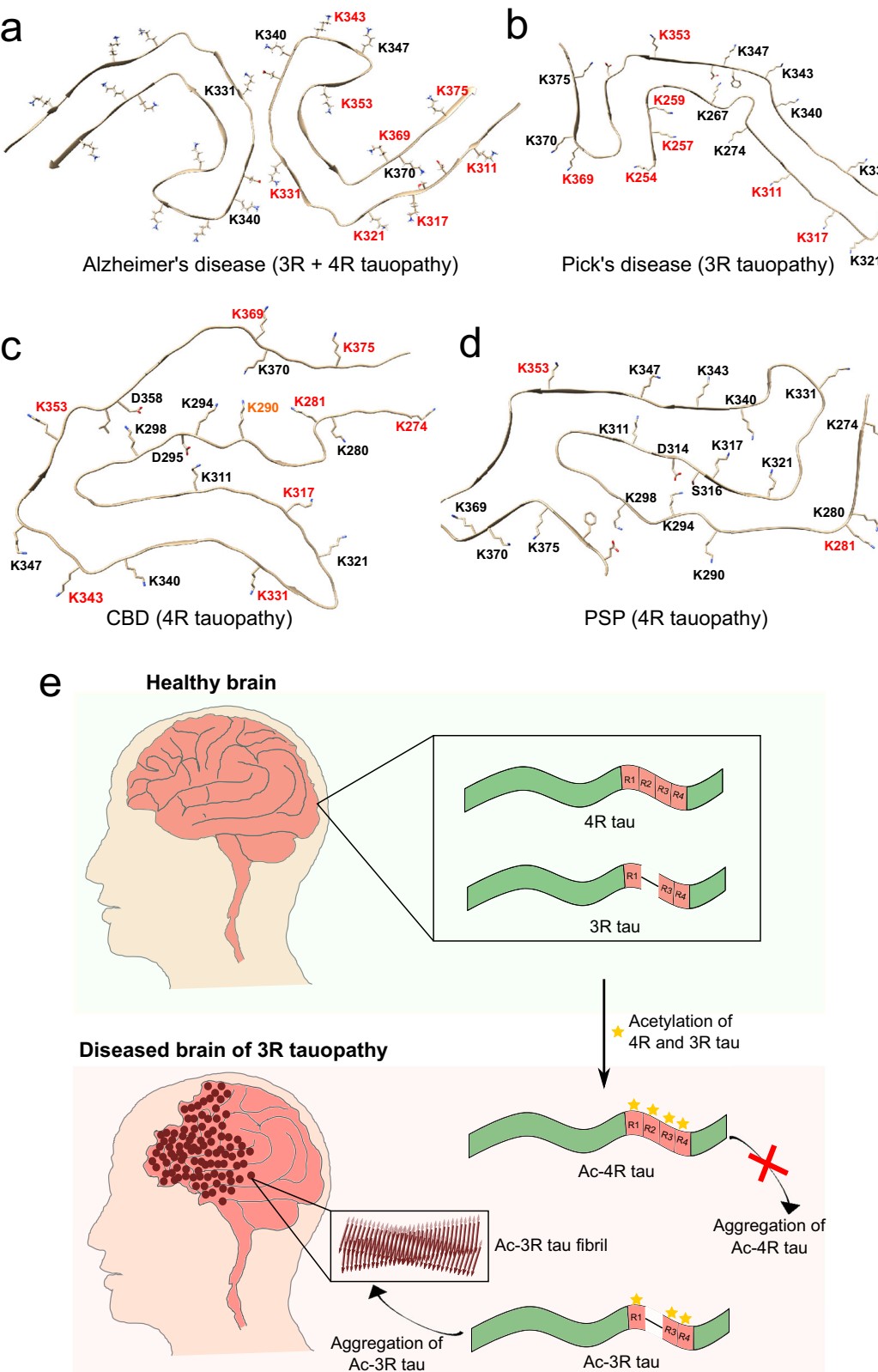

**Fig. 6 | Selective acetylation in tauopathies. a–d** Acetylation patterns mapped onto the atomic structures of tau fibrils purified from Alzheimer's disease patient brain[46] (**a**; PDB code – 5O3L), Pick's disease patient[8] (**b**; PDB code – 6GX5), CBD patient[38] (**c**; PDB code – 6TJO) and PSP patient brain[3] (**d**; PDB code – 7P65). Unmodified lysine residues present in the filament core of each disease are shown in black; acetylated lysine residues in red. Weakly acetylated K290 in CBD is shown in orange. The data of acetylated lysine residues in different tauopathies were taken from Kametani et al.[14] and Arakhamia et al.[13]. **e** Model for the emergence of 3R tauopathies. See text for further details.

Mutants of 2N4R tau were obtained by site-directed mutagenesis using a thermocycler (SensoQuest Labcycler). Phusion high-fidelity PCR master mix (Thermofisher) was used to perform PCR reactions. The sequence of the primers used to generate the mutants are available in Supplementary Table 1.

To prepare unlabeled protein, a single colony from the LB-agar plate was taken and grown overnight in 50 mL LB medium supplemented with 100 µg/mL Ampicillin at 37 °C. 22 mL of the overnight culture were transferred to 1 L LB medium supplemented with 100 µg/mL Ampicillin and allowed to grow until $OD_{600}$ of 0.8–0.9 was reached. Subsequently, the cells were induced with 0.5 mM IPTG (in case of 2N4R tau) or 1 mM IPTG (in case of 0N3R tau) and expressed for 1 h.

To obtain uniformly $^{13}C/^{15}N$-labeled 3R tau, cells were grown in 8 L LB until an $OD_{600}$ of 0.6–0.8 was reached, then centrifuged at low speed ($5000 \times g$), washed with 1X M9 salts, and resuspended in 2 L M9 minimal medium supplemented with 1 g/L $^{15}NH_4Cl$ as the only nitrogen source, 4 g/L $^{13}C$ Glucose as carbon source. After 1 h, the cells were induced with 1 mM IPTG (for 3R tau) and expressed overnight at 37 °C.

To obtain proteins specifically labeled with $^{15}N$ in lysines, cells were grown in 4 L LB until an $OD_{600}$ of 0.6–0.8 was reached, then centrifuged at low speed ($5000 \times g$), washed with 1X M9 salts, resuspended in 1 L M9 minimal medium. After 30 min, 150 mg/L of L-Lysine-2-$^{15}N$-dihydrochloride (Sigma-Aldrich) was added, and after another 30 min, cells were induced with 0.5 mM IPTG (for 4R tau) or 1 mM IPTG (for 3R tau) and expressed overnight at 37 °C.

After harvesting, cell pellets were resuspended in lysis buffer (20 mM MES pH 6.8, 1 mM EGTA, 2 mM DTT) complemented with protease inhibitor mixture, 0.2 mM $MgCl_2$, lysozyme, and DNAse I. Subsequently, cells were disrupted with a French pressure cell press (in ice-cold conditions to avoid protein degradation). NaCl was added to a final concentration of 500 mM, and lysates were boiled for 20 min. Denatured proteins were removed by ultracentrifugation with $127,000 \times g$ at 4 °C for 30 min. To precipitate the DNA, 20 mg/mL streptomycin sulfate was added to the supernatant and incubated for 15 min at 4 °C followed by centrifugation at $15,000 \times g$ for 30 min. The pellet was discarded, and tau protein was precipitated by adding 0.361 g/mL ammonium sulfate to the supernatant, followed by centrifugation at $15,000 \times g$ for 30 min. The pellet containing tau protein was resuspended in buffer A (20 mM MES pH 6.8, 1 mM EDTA, 2 mM DTT, 0.1 mM PMSF, 50 mM NaCl) and dialyzed against the same buffer (buffer A) to remove excess salt. The next day, the sample was filtered and applied to an equilibrated ion-exchange chromatography column (Mono S 10/100 GL, GE Healthcare), and weakly bound proteins were washed out with buffer A. Tau protein was eluted with a linear gradient of 60% final concentration of buffer B (20 mM MES pH 6.8, 1 M NaCl, 1 mM EDTA, 2 mM DTT, 0.1 mM PMSF). Protein samples were concentrated by ultrafiltration (5 kDa Vivaspin, Sartorius) and further purified by reverse phase chromatography using a preparative C4 column (Vydac 214 TP, 5 µm, 8 × 250 mm) in an HPLC system coupled with ESI mass spectrometer. Protein purity was confirmed using mass spectrometry, and the purified protein was lyophilized and re-dissolved in the buffer of interest.

## Aggregation assays
Unmodified wild-type/mutant 4R tau, unmodified 3R tau, acetylated 4R and 3R tau, as well as acetyl-lysine mimics of 4R tau, were aggregated using the co-factor free aggregation protocol[19]. Accordingly, 25 µM of protein were aggregated at 37 °C in 25 mM HEPES, 10 mM KCl, 5 mM $MgCl_2$, 3 mM TCEP, 0.01% $NaN_3$, pH 7.2 buffer (aggregation assay buffer) in a 96-well plate using a Tecan spark plate reader. Three PTFE beads along with double orbital shaking were used to promote fibrilization. Thioflavin-T (ThT) at a final concentration of 50 µM was used to monitor the aggregation kinetics.

## Circular dichroism
10 µL of 25 µM 4R tau fibrils, 3R tau fibrils, and acetylated 3R tau fibrils were pelleted down by centrifugation at $20,000 \times g$ using an Eppendorf centrifuge 5424. The supernatant was discarded, and the pellet was dissolved in 50 µL of distilled water. Acetylated 4R tau was diluted to a final concentration of 5 µM after 5 days of aggregation. CD data were collected in a 0.02 cm pathlength cuvette using Chirascan-plus qCD spectrometer at 25 °C. Datasets were averaged from ten repeated measurements. The spectra were baseline corrected and smoothed with a window size of four.

## Electron microscopy
40 µL of 25 µM unmodified/acetylated 3R tau fibrils were pelleted down by centrifugation at $20,000 \times g$ using an Eppendorf centrifuge 5424. The supernatant was discarded, and the pellet was re-dissolved in 30 µL 25 mM HEPES, 500 mM KCl, 10 mM $MgCl_2$, 3 mM TCEP, 0.01% $NaN_3$, pH 7.2 buffer. Aggregated samples were stained by 1% uranyl acetate solution after adsorbing onto carbon-coated copper grids. The images were taken with a Tietz F416 CMOS camera (TVIPS, Gauting, Germany) using a CM 120 transmission electron microscope (FEI, Eindhoven, The Netherlands).

## Protease digestion
50 µL of 0.8 mg/mL unmodified 3R tau fibrils and 0.4 mg/mL of trypsin (T8003, Sigma-Aldrich), or 0.4 mg/mL of pronase (53702, Merck-Millipore), were incubated in the aggregation assay buffer for 30 min with 1400 rpm shaking in an Eppendorf thermomixer at 37 °C. The trypsin-resistant material was pelleted down by ultracentrifugation at $160,000 \times g$ for 30 min at 4 °C using a Beckman Coulter Optima MAX-UP ultracentrifuge. The supernatant was removed, and the pellet was re-dissolved in 10 µL of aggregation assay buffer and loaded in a 15% SDS-PAGE gel. For mass spectrometry, the tau band from the SDS-PAGE gel was cut and digested by trypsin, followed by detection of the peptides using an ESI mass spectrometer (Orbitrap Fusion Tribrid, Thermo Fischer Scientific)[19].

Pronase digestion of acetylated 3R tau fibrils, as well as 4R tau fibrils aggregated in the presence of acetylated 3R tau seeds, were performed using the same protocol as described above.

## Microtubule assembly
To form microtubule, 25 µM tubulin in BRB80 buffer (100 mM PIPES, 1 mM $MgSO_4$, 1 mM EGTA, 1 mM DTT, pH 6.9) and 1 mM GTP were incubated at 37 °C with 350 rpm shaking for 30 min in an Eppendorf thermomixer. After 30 min, 25 µM paclitaxel (Sigma-Aldrich) was added to the mixture and further incubated for another 30 min at 37 °C with 350 rpm shaking. The suspensions of the sample were fractionated by ultracentrifugation at $160,000 \times g$ for 45 min. The microtubule pellet was resuspended with 25 µM $^{15}N$-lysine labeled 4R tau in the aggregation assay buffer.

## Acetylation of tau
200 µM of 4R tau, 3R tau, $^{15}N$-lysine labeled 4R tau, or $^{15}N$-lysine labeled 3R tau were incubated at 30 °C for 12 h in an Eppendorf thermomixer with 300 rpm shaking in the presence of either 0.028 mg/mL CBP (BML-SE452, Enzo), or 0.028 mg/mL p300 (BML-SE451, Enzo), or both and 20 mM acetyl-coA (Sigma-Aldrich), 1 mM PMSF, 1 mM EGTA. The sample was then boiled at 98 °C for 20 min to precipitate the acetyltransferases, followed by centrifugation at $20,000 \times g$ in an Eppendorf centrifuge 5424. Next, the pellet was discarded, and the supernatant containing acetylated tau was dialyzed against the aggregation assay buffer or NMR buffer (50 mM NaP, 10 mM NaCl, 1 mM TCEP, pH 6.8). Auto-acetylation reactions were performed using the same protocol as described above but only in the absence of acetyltransferases.

To identify the most reactive lysine residues of tau, 200 μM of 4R tau, [15]N-lysine labeled 4R tau were incubated at 30 °C for 2 h in an Eppendorf thermomixer with 300 rpm shaking in the presence of 0.028 mg/mL CBP (BML-SE452, Enzo), 0.028 mg/mL p300 (BML-SE451, Enzo), and 2.5 mM acetyl-coA (Sigma-Aldrich), 1 mM PMSF, 1 mM EGTA. The sample was then boiled at 98 °C for 20 min to precipitate the acetyltransferases, followed by centrifugation at 20,000 × g in an Eppendorf centrifuge 5424. Next, the pellet was discarded, and the supernatant containing acetylated tau was dialyzed against the aggregation assay buffer or NMR buffer (50 mM NaP, 10 mM NaCl, 1 mM TCEP, pH 6.8).

The acetylation reaction of 25 μM microtubule-bound [15]N-lysine labeled tau was performed by incubation at 30 °C for 2 h with 300 rpm shaking in the presence of 0.0035 mg/mL p300 (BML-SE451, Enzo), 0.0035 mg/mL CBP (BML-SE452, Enzo), 2.5 mM acetyl-coA (Sigma-Aldrich), 1 mM PMSF, 1 mM EGTA. Subsequently, 1 M NaCl was added to the solution to break the interaction between microtubule and tau, followed by boiling at 98 °C for 20 min. The precipitated microtubules and enzymes were separated by ultracentrifugation at 160,000 × g for 30 min. Next, the pellet was discarded, and the supernatant containing acetylated tau was dialyzed against the NMR buffer (50 mM NaP, 10 mM NaCl, 1 mM TCEP, pH 6.8). Auto-acetylation reactions in the presence of microtubules were performed using the same protocol as described above but only in the absence of acetyltransferases.

### Preparation of acetyl-lysine mimics

Acetyl-lysine mimics were created using a previously published protocol[31]. At first, 200 μM of C291S/C322S/K280C 4R tau, C291S/C322S/K294C 4R tau, C291S/C322S/K298C 4R tau, C291S/C322S/K311C 4R tau were mixed with 20 mM 2,5-dibromopentanoic acid methyl ester (sc-481713, Chemcruz) in 20 mM NaP, pH 8.0 buffer and incubated at 37 °C for 12 h in an Eppendorf thermomixer with 500 rpm shaking. After the reaction, the excess 2,5-dibromopentanoic acid methyl ester was removed by passing the solution three times through a spin desalting column with 7 KDa MW cut off (Thermo fisher). Next, N-acetylcysteamine (363340, Sigma-Aldrich) was added to the solution to a final concentration of 200 mM, and the mixture was incubated at 37 °C for 12 h in an Eppendorf thermomixer with 500 rpm shaking. After 12 h, the excess N-acetylcysteamine was removed by passing the solution three times through a spin desalting column with 7 KDa MW cut off (Thermo Fisher) followed by dialysis against the aggregation assay buffer. The increase in mass by 85 Da confirmed the formation of the acetyl-lysine mimics (Supplementary Fig. 8).

### In-gel digestion and extraction of peptides for mass spectrometry

The respective bands from the SDS-PAGE gels were carefully cut and kept in an Eppendorf tube. To wash the gel pieces, 150 μL of water was added and incubated for 5 min at 26 °C with 1050 rpm shaking in a thermomixer. The gel pieces were spun down and the liquid was removed using thin tips (the same washing protocol was used in all subsequent steps with different solvents). The gel pieces were washed again with 150 μL acetonitrile. After washing, the gel pieces were dried for 5 min using a SpeedVacc vacuum centrifuge. To reduce disulfide bridges, 100 μL of 10 mM DTT was added to the gel pieces and incubated for 50 min at 56 °C followed by centrifugation and removal of liquid. The gel pieces were washed again with 150 μL of acetonitrile. To alkylate reduced cysteine residues, 100 μL of 55 mM iodoacetamide were added and incubated for 20 min at 26 °C with 1050 rpm shaking followed by centrifugation and removal of liquid. Subsequently, the gel pieces were washed with 150 μL of 100 mM NH4HCO3, and then twice with 150 μL of acetonitrile and dried for 10 min in a vacuum centrifuge. The gel pieces were rehydrated at 4 °C for 45 min by addition of small amounts (2–5 μL) of digestion buffer 1 (12.5 μg/mL trypsin, 42 mM NH4HCO3, 4 mM CaCl2). The samples were checked after every 15 min and more buffer was added in case the liquid was completely absorbed by the gel pieces. 20 μL of digestion buffer 2 (42 mM NH4HCO3, 4 mM CaCl2) were added to cover the gel pieces and incubated overnight at 37 °C.

To extract the peptides, 15 μl water was added to the digest and incubated for 15 min at 37 °C with 1050 rpm shaking followed by spinning down the gel pieces. 50 μl acetonitrile was added to the entire mixture and incubated for 15 min at 37 °C with 1050 rpm shaking. The gel pieces were spun down and the supernatant (SN1) containing the extracted peptides was collected. 30 μl of 5% (v/v) formic acid was added to the gel pieces and incubated for 15 min at 37 °C with 1050 rpm shaking followed by spinning down. Again 50 μl acetonitrile were added to the entire mixture and incubated for 15 min at 37 °C with 1050 rpm shaking. The gel pieces were spun down and the supernatant (SN2) containing the extracted peptides was collected. Both supernatants (SN1 & SN2) containing the extracted peptides were pooled together and evaporated in the SpeedVacc vacuum centrifuge. The dried peptides were resuspended in 5% acetonitrile and 0.1% formic acid and analyzed using an Orbitrap Fusion Tribrid (Thermo Fischer Scientific) instrument. The MS data were analyzed using Scaffold 4 software.

### NMR spectroscopy

Two-dimensional [1]H-[15]N HSQC spectra of [13]C/[15]N labeled 3R tau monomer were acquired at 278 K in the aggregation assay buffer on a Bruker Avance III 900 MHz spectrometer equipped with a 5 mm HCN cryoprobe. Chemical shift assignment was performed by transferring the assignment of 4R tau[41] and helped with a previous assignment of 3R Tau at 25 °C[42] (BMRB 50701).

To record the [1]H-[15]N J-transfer MAS spectra, [13]C/[15]N labeled 3R tau was aggregated without ThT using the protocol described above, and ~30 mg of fibrils were packed into a 3.2 mm MAS rotor. The proton-detected [1]H-[15]N J-transfer spectra were acquired at 265 K and 17 kHz MAS on an Avance III 950 MHz spectrometer (Bruker) using a 3.2 mm E-free HCN probe with 32 scans per point (ns), and indirect acquisition times td1 = 36 ms, td2 = 40 ms.

Signals in the [1]H-[15]N J-transfer spectra of 3R tau fibrils were assigned by transferring the resonance assignment of the 3R tau monomer. Intensity ratios were calculated by dividing the intensity of each residue in the [1]H-[15]N J-transfer spectrum of the fibril sample by the intensity observed in the [1]H-[15]N HSQC spectrum of the monomeric protein. The residue with the highest intensity was normalized to 1.0, and the average line was calculated by smoothening using a 2nd order polynomial function with window size four.

The [1]H-[15]N HSQC spectra of 50 μM [15]N-lysine labeled unmodified/acetylated 4R/3R tau or [15]N labeled unmodified/acetylated 4R tau were recorded at 278 K in 50 mM NaP, 10 mM NaCl, 1 mM TCEP, pH 6.8 buffer using an Avance III 900 MHz spectrometer (Bruker) equipped with a 5 mm HCN cryoprobe and an Avance neo 800 MHz spectrometer (Bruker) equipped with a 3 mm HCN cryoprobe. The signals in the [1]H-[15]N HSQC spectra of [15]N-lysine labeled unmodified 4R and 3R tau were assigned by transferring the assignment from the [1]H-[15]N HSQC spectra of uniformly [15]N-labeled 4R and 3R tau[41]. Some peaks of the acetylated 4R tau were assigned according to previously published studies[43]. The acetylation level of individual lysine residues was determined using the following equation:

$$\text{Acetylation level}\,(\%)$$
$$= \left\{ 1 - \frac{\text{Intensity of unacetylated lysine (in the acetylated sample)}}{\text{Intensity of unmodified lysine (in the non-acetylated sample)}} \right\} \times 100$$

$$(1)$$

The known assignments of the resonances of non-acetylated lysine residues affected due to the acetylation of neighboring residue were taken into account while determining the acetylation levels: for

example, while determining the acetylation level of K298, we added the intensities of the unmodified K298 peak and the resonance of unmodified K298 affected by the Ac-K294.

[1]H-detected hNH[44] and hCANH[44] experiments were recorded on a Bruker Avance III 850 MHz spectrometer, at 55 KHz MAS using a triple-resonance HCND 1.3 mm probe. The VT gas temperature was set at 240 K (sample temperature ~290 K). Approximately 3 mg of 2N4R and 1.75 mg of 2N4R acetylated fibrils were packed into 1.3 mm MAS rotors by ultracentrifugation. The 90° pulses were set to 2.5 µs at 15 watts for [1]H, 5 µs at 60 watts for [15]N, and 5 µs at 15 watts for [13]C. Two-dimensional hNH experiments of both fibril samples were recorded with 32 scans per point (ns), and indirect acquisition times td1 = 19.14 ms, td2 = 39.5 ms. Each three-dimensional hCANH experiments were recorded for ~21 h with td1 = 10.9 ms, td2 = 8.5 ms, td3 = 19.7 ms. The final spectra for 2N4R acetylated and 2N4R fibrils are the sum of 3 and 2 datasets for hNH experiments, and 6 and 3 datasets for hCANH. Spectra were processed in Topspin v3.6.1 (Bruker) and analyzed with CcpNmr-Analysis v2.4.2[45].

### Reporting summary
Further information on research design is available in the Nature Portfolio Reporting Summary linked to this article.

## Data availability
All the PDB codes cited in this paper (5O3L, 6GX5, 6TJO, 7P65) are available in the protein data bank web server. NMR assignments referenced in this study can be accessed using the following accession code: BMRB 50701. Source data are provided with this paper.

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

## Acknowledgements

We thank the mass spectrometry facility of the Max Planck Institute for Multidisciplinary Sciences (MPI-NAT, Göttingen) for mass spectrometry and the EM facility of MPI-NAT for electron micrographs. We thank Maria-Sol Cima-Omori, DZNE Göttingen, for helping in the preparation of some constructs of 2N4R tau. We thank Kerstin Overkamp, MPI-NAT, for purifying tau constructs by reverse-phase chromatography. M.Z. was supported by the European Research Council (ERC) under the EU Horizon 2020 research and innovation program (grant agreement No. 787679).

## Author contributions

P.C. prepared tau constructs, conducted protein purification, enzymatic as well as specific acetylation reactions, aggregation assays, protease digestion experiments, microtubule polymerization, biophysical analysis, and solution-state NMR experiments; G.R. conducted solid-state NMR experiments; A.H. performed experiments and contributed to discussions; A.I.O. supported biophysical analysis; I.M.V. supervised experiments and contributed to discussions; L.B.A. supported solid-state NMR experiments; M.Z. supervised the project; P.C. and M.Z. designed the project and wrote the paper.

## Funding

## Competing interests

The authors declare no competing interests.
