## [Peer Review File · Nature Communications]

Reviewers' Comments:

Reviewer #1:

Remarks to the Author:

In this paper, Chakraborty et al describe the isoform-specific aggregation of Tau protein by lysine acetylation by CBP/p300 acetyltransferases. Acetylation promotes self-assembly of a 3R tau isoform into amyloid fibrils while preventing aggregation of a 4R isoform that even resist seeding with acetylated 3R tau fibrils. Furthermore, the role of lysine K298 acetylation is highlighted in the aggregation process using acetylation mimics and pseudo-acetylation. Structure of the 3R tau fibril core either in its non-acetylated or acetylated state is further examined by solid-state NMR (ssNMR) showing a difference in fibrillar structures upon site-specific acetylation.

Overall, the paper relates to a highly relevant issue in the field of neurodegenerative diseases and the claims are appealing. However, the conclusions of the manuscript are mainly based on the absence of aggregation of the acetylated 4R tau isoform which would require a strong validation given the difficulty in triggering tau aggregation without any co-factor and reproducibility issues. Moreover, the ssNMR investigation of the acetylation-induced structural changes in 3R tau fibrils is scarce.

More specifically, the acetylation levels obtained with CBP or p300 or both acetyltransferases are strikingly low for most of lysine residues except those surrounding cysteine residues (C291 and C322). The pattern described here is significantly different to what has been previously described by NMR spectroscopy for acetylated 4R tau and seems rather non-specific. It looks like acetylation is rather provided by an autoacetylation process that is probably favored by the high concentration of acetyl-coenzyme A used in the acetylation reactions. It would be worth examining the activity of acetyltransferases on a standard peptide substrate and in parallel, the acetylation of tau in the same conditions used here without enzyme. Otherwise, mutation of native cysteine residues could significantly reduce autoacetylation. Hence, the conclusions about the role of enzymatic acetylation on the fibrillar aggregation of tau should be taken with caution.

Moreover, the estimation of site-specific acetylation levels can be put into question. Determining site-specific modification levels of less than 5% using NMR signal intensities seems to be cryptic (e.g. K44, K67, K87, K130, etc...). Here, the site-specific acetylation levels were determined by comparing the intensities of each lysine signal in the acetylated vs. non-acetylated sample requiring the precise determination of protein concentration/normalization in both samples. This method is indirect as it is not based on a direct measurement of the signal intensity of acetylated lysines. It provides wrong results in the case of residues that are very close in the protein sequence such as K224/K225, K280/K281, K369/K370, K294/K298, etc... as the proximity of acetylation sites leads to splitting of resonances into more than two peaks for every lysine involved. In this case, the intensity loss of the signal corresponding to the non-acetylated form only is inappropriate to provide the acetylation level. This issue should be clarified.

Amyloid fibrillar aggregation is adequately performed in the absence of any external inducer. The effect of acetylation on the fibrillation of tau 3R and 4R isoforms is investigated using standard techniques such as ThT fluorescence emission and electron microscopy as well as limited proteolysis of aggregated species with pronase. Despite an increase in ThT signal for the 3R isoform in its acetylated and non-acetylated form and the non-acetylated 4R isoform, the formation of amyloid-like fibrils is not convincing based on TEM images presented which are of poor quality. Of note, the negative staining shows fibrils that are poorly contrasted, especially in Fig.3. The kinetics curves of aggregation correspond to three independent experiments from the same (acetylated or non-acetylated) sample but it would be worth measuring the aggregation kinetics of different acetylated/non-acetylated samples. The complete lack of aggregation of acetylated 4R tau must be confirmed using independent acetylated samples as it is a crucial point in the conclusion drawn in the manuscript. Given the high sensitivity of tau aggregation to subtle changes in the conditions of aggregation, such a total loss of aggregation ability should be carefully checked. The same is true for the seeding experiments as it most probably used the same acetylated 4R tau sample that is unable to aggregate. In contrast, acetylated 3R tau fibrils are able to seed aggregation of a 4R tau monomer that has the self-capacity to form fibrils without seeds excluding cross-seeding issues.

Finally, the ssNMR data indicate structural changes in the acetylated vs. non-acetylated fibrils. In the absence of bona fide resonance assignment, variations of NMR signals could be simply due to the effect of acetylation on resonances of lysine and neighboring residues even in the ^{13}C - ^{15}N planes: the spectral heterogeneity could originate from the heterogeneity in the acetylation proteoforms. The interpretation of spectral changes induced by acetylation in terms of structural effect requires a detailed assignment of resonances.

Minor comments:

- the authors should explain why they are using 0N3R tau as 3R isoform and 2N4R tau as 4R isoform.

Reviewer #2:

Remarks to the Author:

This manuscript reports the interesting observation that acetylation changes the fibrillization kinetics of 4R and 3R tau proteins in opposite directions. P300/CBP based acetylation across full-length proteins accelerated 3R tau fibrillization but retarded 4R tau fibrillization. Site-specific Lys to Gln mutations reproduced this trend. The authors interpret this differential effect as the potential reason for the enrichment of 3R tau aggregates over 4R tau aggregates in the 3R tauopathy of Pick's disease. While the results are interesting, many data are superficially analyzed and some interpretations are speculative. The authors should revise the manuscript to address the following questions:

1. The 3R tau data indicate that R3 acetylation accelerates fibril growth. So why does the abundant acetylation of R3 residues in 4R tau not counteract the slowing down effect of R2 acetylation?
2. Please explain why the pattern of acetylation level is so different between the MT-bound 4R tau (Figure 2d) and 4R tau alone (Figure S2). How do P300/CBP interact with microtubules?
3. It is peculiar that the N-terminal 30–40 residues of the protein are immobilized in the unmodified 3R tau fibrils. The authors attribute this to the existence of an epitope in this region for mAbs. This is not convincing, as neither Pick's disease tau nor other tauopathy tau fibrils have been shown to immobilize the N-terminal region. Are unmodified 4R tau fibrils also missing these N-terminal residues?
4. Please add the MAS-HSQC ^1H - ^{15}N 2D spectra of acetylated 3R tau fibrils, and compare the peak intensities with the 3R tau monomers. How do the intensity ratios differ from those of the unmodified 3R tau fibrils?
5. Given the general difficulty of cross seeding, it is not surprising to this reviewer that acetylated 3R tau does not seed acetylated 4R tau, while it is surprising that acetylated 3R tau accelerated fibril formation of unmodified 4R tau. Please add the TEM data of this cross-seeded sample, and compare the MAS-HSQC spectrum of the acetyl-3R seeded 4R tau sample with the unmodified 4R tau fibrils.
6. Are there biological data in the literature about the acetylation levels in Pick's disease, in 4R tauopathies, and in AD? This biological data would be very useful for verifying the interpretation of the manuscript.
7. Although it's understandable and tempting to rationalize the impact of specific Lys residues' acetylation on fibrillization kinetics in terms of ex vivo brain tau fibril structures (Figure 6), there is no data in this paper that the fibrils prepared here are the same as the ex vivo brain tau. In fact more likely, these in vitro fibrils are somewhat different from the ex vivo fibrils. So the authors should revise the discussion to clarify that their interpretations are only rationalization and models, to be tested by experiments in the future.

Reviewer #3:
None

Reviewer #1:

In this paper, Chakraborty et al describe the isoform-specific aggregation of Tau protein by lysine acetylation by CBP/p300 acetyltransferases. Acetylation promotes self-assembly of a 3R tau isoform into amyloid fibrils while preventing aggregation of a 4R isoform that even resist seeding with acetylated 3R tau fibrils. Furthermore, the role of lysine K298 acetylation is highlighted in the aggregation process using acetylation mimics and pseudo-acetylation. Structure of the 3R tau fibril core either in its non-acetylated or acetylated state is further examined by solid-state NMR (ssNMR) showing a difference in fibrillar structures upon site-specific acetylation. Overall, the paper relates to a highly relevant issue in the field of neurodegenerative diseases and the claims are appealing.

Reply: We thank the reviewer for the careful evaluation of the manuscript and highlighting the relevance of the study.

However, the conclusions of the manuscript are mainly based on the absence of aggregation of the acetylated 4R tau isoform which would require a strong validation given the difficulty in triggering tau aggregation without any co-factor and reproducibility issues. Moreover, the ssNMR investigation of the acetylation-induced structural changes in 3R tau fibrils is scarce. More specifically, the acetylation levels obtained with CBP or p300 or both acetyltransferases are strikingly low for most of lysine residues except those surrounding cysteine residues (C291 and C322). The pattern described here is significantly different to what has been previously described by NMR spectroscopy for acetylated 4R tau and seems rather non-specific. It looks like acetylation is rather provided by an autoacetylation process that is probably favored by the high concentration of acetyl-coenzyme A used in the acetylation reactions. It would be worth examining the activity of acetyltransferases on a standard peptide substrate and in parallel, the acetylation of tau in the same conditions used here without enzyme. Otherwise, mutation of native cysteine residues could significantly reduce autoacetylation. Hence, the conclusions about the role of enzymatic acetylation on the fibrillar aggregation of tau should be taken with caution.

Reply: We thank the reviewer for the thoughtful comment. As suggested by the reviewer, we performed the acetylation reaction of both 4R and 3R tau in the absence of acetyltransferases and determined the acetylation levels by NMR. Under the condition of auto-acetylation, the lysine residues nearby to the two cysteine residues (C291, C322) of tau reached a comparable level of acetylation as of the acetylation level in the presence of acetyltransferases (Supplementary Fig. 2a,c). This suggests that the strong acetylation of the five lysine residues of repeat R2 was mostly due to auto-acetylation of tau. However, the lysine residues away from the cysteine residues of tau reached a higher degree of acetylation in the presence of acetyltransferases (Supplementary Fig. 2b,d). The results are described on Page no. 7 of the revised manuscript.

To identify the most reactive lysine residues of tau, we acetylated 4R tau in the presence of both P300 and CBP acetyltransferases for two hours. We also decreased the concentration of acetyl co-A to 2.5 mM. The reduction of reaction time from twelve hours to two hours leads to a global decrease in acetylation levels (Supplementary Fig. 3a). However, the lysine residues nearby to the two cysteine residues of tau (C291, C322) were efficiently acetylated.

To study how the duration of the acetylation reaction, as well as the acetylation levels of individual lysine residues, impacts the aggregation of 4R tau, we aggregated the 4R tau, which was acetylated for two hours in the presence of P300 and CBP acetyltransferases. The two-hour acetylation also strongly delayed the aggregation of 4R tau (Supplementary Fig. 3b,c) and lead to the aggregation of only ~20 % acetylated protein (Supplementary Fig. 3d,e). This is in line with our findings that prolonged acetylation (twelve hours) of 4R tau inhibits its aggregation (Fig. 3a). The results are described on Page no. 7 & 10 of the revised manuscript.

Moreover, the estimation of site-specific acetylation levels can be put into question. Determining site-specific modification levels of less than 5% using NMR signal intensities seems to be cryptic (e.g. K44, K67, K87, K130, etc...). Here, the site-specific acetylation levels were determined by comparing the intensities of each lysine signal in the acetylated vs. non-acetylated sample requiring the precise determination of protein concentration/normalization in both samples. This method is indirect as it is not based on a direct measurement of the signal intensity of acetylated lysines. It provides wrong results in the case of residues that are very close in the protein sequence such as K224/K225, K280/K281, K369/K370, K294/K298, etc... as the proximity of acetylation sites leads to splitting of resonances into more than two peaks for every lysine involved. In this case, the intensity loss of the signal corresponding to the non-acetylated form only is inappropriate to provide the acetylation level. This issue should be clarified.

Reply: Thanks for allowing us to clarify this point. We performed precise normalization of the 2D spectra using the 1D ¹H spectra of the unmodified and acetylated K-labeled tau samples before determining the acetylation levels. We agree with the reviewer that for certain K residues that are close in sequence (K224/K225, K280/K281, K369/K370), the determination of acetylation level is not precise due to the splitting of the resonances. We acknowledge this limitation in the revised manuscript on page no. 5. In addition, some lysine residues were excluded from the analysis due to the overlap of the acetylated lysine resonance with other resonances of tau. We also used the known assignments of the resonances of non-acetylated lysine residues affected due to the acetylation of neighboring residue while determining the acetylation levels: for example, while determining the acetylation level of K298, we added the intensities of the unmodified K298 peak and the resonance of unmodified K298 affected by the Ac-K294. This is now stated on page 24 of the revised manuscript.

Amyloid fibrillar aggregation is adequately performed in the absence of any external inducer. The effect of acetylation on the fibrillation of tau 3R and 4R isoforms is investigated using standard techniques such as ThT fluorescence emission and electron microscopy as well as limited proteolysis of aggregated species with pronase. Despite an increase in ThT signal for the 3R isoform in its acetylated and non-acetylated form and the non-acetylated 4R isoform, the formation of amyloid-like fibrils is not convincing based on TEM images presented which are of poor quality. Of note, the negative staining shows fibrils that are poorly contrasted, especially in Fig.3.

Reply: We have updated the negative-stain electron micrograph of the acetylated 3R tau fibril in Fig. 3e.

The kinetics curves of aggregation correspond to three independent experiments from the same (acetylated or non-acetylated) sample but it would be worth measuring the aggregation kinetics of different acetylated/non-acetylated samples. The complete lack of aggregation of acetylated 4R tau must be confirmed using independent acetylated samples as it is a crucial point in the conclusion drawn in the manuscript. Given the high sensitivity of tau aggregation to subtle changes in the conditions of aggregation, such a total loss of aggregation ability should be carefully checked. The same is true for the seeding experiments as it most probably used the same acetylated 4R tau sample that is unable to aggregate. In contrast, acetylated 3R tau fibrils are able to seed aggregation of a 4R tau monomer that has the self-capacity to form fibrils without seeds excluding cross-seeding issues.

Reply: The ThT fluorescence experiment of both acetylated 3R and 4R tau has been performed up to 5 times with 3 different batches of tau. Before performing each of the ThT fluorescence experiments, the acetylation reaction has been performed independently. In all cases the data were reproducible. This information is provided under the “replication” section of the Nature reporting summary and has now been added to the legend of Fig. 3a.

Finally, the ssNMR data indicate structural changes in the acetylated vs. non-acetylated fibrils. In the absence of bona fide resonance assignment, variations of NMR signals could be simply due to the effect of acetylation on resonances of lysine and neighboring residues even in the ^{13}C - ^{15}N planes: the spectral heterogeneity could originate from the heterogeneity in the acetylation proteoforms. The interpretation of spectral changes induced by acetylation in terms of structural effect requires a detailed assignment of resonances.

Reply: We recorded 2D hNH, 3D (H)CANH, and also 2D NCA, 3D NCACX, 3D NCOCX spectra to assign the unmodified/acetylated 3R tau fibrils. Unfortunately, we were unable to assign the fibrils due to strong overlap of the signals as well as low signal intensities in the 3D experiments. We added the following sentence on the page no. 15 of the revised manuscript –

“However, as we lack the residue-specific resonance assignments, variation of the NMR signals in the acetylated state might also be simply due to the effect of acetylation on the resonance of lysine and neighboring residues.”

Minor comments:

- the authors should explain why they are using ON3R tau as 3R isoform and 2N4R tau as 4R isoform.

Reply: The rationale behind choosing the isoforms was to work with the two extreme isoforms of tau, i.e., the shortest construct ON3R tau, and the longest construct 2N4R tau.

Reviewer #2:

This manuscript reports the interesting observation that acetylation changes the fibrillization kinetics of 4R and 3R tau proteins in opposite directions. P300/CBP based acetylation across full-length proteins accelerated 3R tau fibrillization but retarded 4R tau fibrillization. Site-specific Lys to Gln mutations reproduced this trend. The authors interpret this differential effect as the potential reason for the enrichment of 3R tau aggregates over 4R tau aggregates in the 3R tauopathy of Pick's disease. While the results are interesting, many data are superficially analyzed and some interpretations are speculative. The authors should revise the manuscript to address the following questions:

Reply: We thank the reviewer for the careful evaluation of the manuscript and highlighting the relevance of the study.

1. The 3R tau data indicate that R3 acetylation accelerates fibril growth. So why does the abundant acetylation of R3 residues in 4R tau not counteract the slowing down effect of R2 acetylation?

Reply: We can rationalize the faster aggregation of 3R tau upon acetylation when considering the importance of electrostatic interactions in tau aggregation: acetylation of lysine residues removes positive charges which decrease electrostatic repulsion and favors intermolecular interactions of the repeat region during tau aggregation. However, simple changes in electrostatics would also predict faster aggregation of acetylated 4R tau, in striking contrast to the observed inhibition of 4R tau aggregation by acetylation (Fig. 3a). This is because the inhibition of aggregation upon acetylation of 4R tau is due to the inability of the acetylated 4R tau monomer to adopt a stable protofilament fold due to the steric crowding between the acetylated lysine residues present in the R2 repeat.

This is discussed in more detail in the discussion section on pages 17 & 18:

“Using site-specific pseudo-acetylation, we showed that acetylation of K298 most strongly delays aggregation of 4R tau (Fig. 4d). Inspection of the 3D structure of tau filaments purified from the 4R tauopathy CBD provides a rationale for the importance of K298 acetylation in discriminating 4R and

3R tauopathies (Fig. 6c). In the tau CBD structure^{13,40}, the side chain of K298 points inside the core of the tau fibrils and forms a salt bridge with D358. When K298 is acetylated the salt bridge can no longer form. In addition, the increased bulkiness of the side chain of K298 upon acetylation will generate steric clash, i.e., 4R tau will be unable to form the CBD structure when K298 is acetylated. The impact of acetylation at K294 on 4R tau aggregation can further be rationalized on the basis of the structure of tau fibrils purified from the 4R tauopathy Progressive Supranuclear Palsy (PSP; Fig. 6d): K294 forms a salt bridge with D314/S316 in the core of the tau PSP structure^{3,41}. Thus, the delay observed in the aggregation of 4R tau upon specifically acetylating K298 or K294 likely arises from the increase in steric crowding in the vicinity of these residues upon acetylation. Thus, acetylation of specific lysine residues in repeat R2 favors the accumulation of 3R tau.

Besides acetylation in repeat R2, acetylation of other lysine residues may contribute to isoform-specific tau deposition¹⁴. For example, K311 is acetylated in the 3R tauopathy Pick's disease, whereas it is not acetylated in the case of the 4R tauopathies CBD and PSP^{14,28}. In the structure of tau fibrils from Pick's disease⁸, K311 points away from the filament core (Fig. 6b). Acetylation of K311 thus will not interfere with the structure of the tau fibril core in Pick's disease. In contrast, K311 forms a salt bridge with D295 in the CBD tau structure (Fig. 6c), and is located in a compact region of the PSP tau structure (Fig. 6d). Selective acetylation of K311 in Pick's disease is thus likely, because acetylation of K311 may interfere with the formation of CBD/PSP tau fibril structures due to steric hindrance. In agreement with this hypothesis, we observed strong inhibition of 4R tau aggregation upon selectively acetylating K311 (Fig. 4d). In addition, selective acetylation of K311 in Pick's disease will be favored by its solvent accessible location in the structure of the Pick's disease fibrils, whereas the K311 side chain is not accessible in the structure of tau fibrils from CBD and PSP (Fig. 6c,d)."

2. Please explain why the pattern of acetylation level is so different between the MT-bound 4R tau (Figure 2d) and 4R tau alone (Figure S2). How do P300/CBP interact with microtubules?

Reply: The acetylation reaction of 4R tau in the absence of MT (Figure S2) was performed for 12 hours in the presence of 20 mM acetyl-coA, i.e., in the condition of abundant acetylation. However, in the case of MT-bound 4R tau, the acetylation reaction was performed only for 2 hours in the presence of 2.5 mM acetyl-coA, i.e., in the condition to acetylate predominantly the most reactive lysine residues. In the MT-bound state, it was important to acetylate only the most-reactive lysine residues because the acetylation decreases the binding affinity of tau to microtubules (ref #36) and we wanted to avoid the complete detachment of tau from microtubules during the course of the acetylation reaction. Unfortunately, the interaction between the p300/CBP and microtubules is not known so far.

3. It is peculiar that the N-terminal 30–40 residues of the protein are immobilized in the unmodified 3R tau fibrils. The authors attribute this to the existence of an epitope in this region for mAbs. This is not convincing, as neither Pick's disease tau nor other tauopathy tau fibrils have been shown to immobilize the N-terminal region. Are unmodified 4R tau fibrils also missing these N-terminal residues?

Reply: Please note that different conformation-specific antibodies (e.g. Alz50, MC1) specifically detect pathological tau in the brain tissue (Ref #20). These antibodies require two discontinuous epitopes, one located in the repeat domain (residue 313-322) and the other at the N-terminus (residue 1-18). This suggests that the N-terminal residues somehow interact with the cross- β -structure of tau fibrils thereby generating the pathology-specific recognition motif.

In the cryo-EM structures of the ex-vivo tau fibrils from different tauopathies, however, the N-terminal residues are not yet detected. This is most probably because the interaction between the N-terminal residues and the fibril core is dynamic and it might thus be lost during the extraction of the filaments from the brain homogenates.

Yes, in the case of unmodified 4R tau fibrils, we also see the immobilization of the N-terminal residues (please see Ref #19).

4. Please add the MAS-HSQC 1H-15N 2D spectra of acetylated 3R tau fibrils, and compare the peak intensities with the 3R tau monomers. How do the intensity ratios differ from those of the unmodified 3R tau fibrils?

Reply: Please note that the motivation behind the recording of the MAS-HSQC 1H-15N 2D spectra of the unmodified 3R tau fibrils was to determine the rigid core of the fibril. We further confirmed the location of the rigid core of unmodified 3R tau fibrils by performing the pronase digestion and analyzing the pronase-resistant core by MS (Fig. 1g). Both the NMR experiment, as well as the MS analysis, provided similar information regarding the rigid core of the unmodified 3R tau fibrils. Also, to record the MAS-HSQC 1H-15N 2D spectra of the unmodified 3R tau fibrils, we used a 3.2 mm rotor that needs around 30 mg of fibril sample.

In the case of the acetylated-3R tau fibrils, we were unable to prepare a fibril sample to fill a 3.2 mm rotor (needs around 30 mg of sample) because of the requirement of a large amount of acetyltransferases. So, we tried to use a 1.3 mm rotor that needs much less sample (around 3 mg). However, because of the design of the 1.3 mm rotor (caps at the top as well as at the bottom), we noticed a water retention problem and the fibril sample was not hydrated any more. Because we detect the dynamic residues present in the sample in the MAS-HSQC experiments, the sample must be sufficiently hydrated to detect the signals of the dynamic residues. We were therefore unable to record the MAS-HSQC 1H-15N 2D spectrum of the acetylated 3R tau fibrils even with a 1.3 mm rotor.

In order to determine the rigid core of the acetylated 3R tau fibril we performed pronase digestion, followed by pelleting down the pronase-resistant core (Fig. 3f). Mass spectrometry of the pronase-resistant band revealed that the acetylated 3R tau fibrils have similar residues in the rigid core as unmodified 3R tau fibrils.

5. Given the general difficulty of cross seeding, it is not surprising to this reviewer that acetylated 3R tau does not seed acetylated 4R tau, while it is surprising that acetylated 3R tau accelerated fibril formation of unmodified 4R tau. Please add the TEM data of this cross-seeded sample, and compare the MAS-HSQC spectrum of the acetyl-3R seeded 4R tau sample with the unmodified 4R tau fibrils.

Reply: We added the TEM data of the cross-seeded fibril as Supplementary Fig. 4b.

To compare the rigid core of the acetyl-3R seeded 4R tau sample with the unmodified 4R tau fibrils we performed the pronase digestion experiment. Upon analyzing the pronase-resistant core by MS we observed that these cross-seeded fibrils (unmodified 4R tau aggregated in the presence of acetylated 3R tau seeds) comprise a longer rigid-core as compared to the rigid-core of unmodified 4R tau fibrils (Supplementary Fig. 4c,d). We added this result to the revised manuscript on page 11.

6. Are there biological data in the literature about the acetylation levels in Pick's disease, in 4R tauopathies, and in AD? This biological data would be very useful for verifying the interpretation of the manuscript.

Reply: Yes, there is information about the residues of tau that are acetylated in Pick's disease, AD, and other 4R tauopathies. Please see Fig. 6a-d and the discussion section on pages 17, 18.

7. Although it's understandable and tempting to rationalize the impact of specific Lys residues' acetylation on fibrillization kinetics in terms of ex vivo brain tau fibril structures (Figure 6), there is no data in this paper that the fibrils prepared here are the same as the ex vivo brain tau. In fact more likely, these in vitro fibrils are somewhat different from the ex vivo fibrils. So the authors should revise the discussion to clarify that their interpretations are only rationalization and models, to be tested by experiments in the future.

Reply: Thanks for this suggestion. We added the following sentence to the discussion section on page 19 of the revised manuscript:

“As we were unable to determine the structure of the in vitro acetylated 3R tau fibrils, the interpretations are only rationalizations and models, to be tested by future experiments.”

Reviewers' Comments:

Reviewer #1:

Remarks to the Author:

The manuscript of Chakraborty et al has been revised substantially according to my comments.

The new experiments that have been done highlight a role of auto-acetylation by significantly modulating the acetylation pattern of acetyltransferases CBP/p300. Notably, the major difference due to auto-acetylation resides in the acetylation levels of lysines in the R2 repeat which is probably the main reason why acetylated tau3R and tau4R behave differently in the aggregation reactions.

In whole, all lysine residues appear to be acetylated (without any exception) at relatively low levels (less than 25%) except those of the R2/R3 repeats around the cysteine residues that display high acetylation levels. Moreover, the autoacetylation levels are comparable if not higher than enzymatic levels in most places which indicates the auto-acetylation is the main driver of acetylation here (see Fig 2d). My opinion is that the determination of the acetylation levels on NMR spectra by comparison of relative intensities between acetylated vs. non-acetylated samples is inappropriate as it clearly appears by looking directly on the spectra of Fig 2b,c that there are not so many acetylation sites. The intensity changes btw. Ac vs. non-Ac tau could be explained by conformational changes and/or differential water exchange of amide protons. This should be clarified by throughout investigation and above all, assigning the acetylation sites (or compare with published data showing the assignment of acetylated Tau spectrum) should provide a direct quantification of acetylation levels.

Another point concerns the acetylation mimics. The lysine-to-glutamine mutation is generally used to mimic the acetylation state (in part by neutralization of the positive charge), e. g. as shown by Min et al., *Nature Medicine* (2015) 21: 1154–1162 or Ajit et al, *JBC* (2019) 294(45):16698-16711. However, the K-to-Q mutants (5 out of 6 mutants) aggregate faster than wild-type 4R tau while acetylation was shown to delay 4R tau aggregation as also illustrated by pseudo-acetylation mimics. This discrepancy should be clearly explained.

Based on the data presented here, I cannot definitely see the benefits of ssNMR on the structural description of acetylated tau fibrils. Given the limited information provided by ssNMR experiments, the "effect of acetylation on the structure of 3R tau fibrils" as indicated in the subtitle is largely speculative. The pronase digestion profile and MS analyses are more convincing. A re-organization of the manuscript on this specific point must be done putting all these data together.

Reviewer #2:

Remarks to the Author:

The authors have addressed most of my original comments. Overall, the manuscript reports many interesting observations about how in vitro acetylation levels vary across the Lys residues, and how acetylation changes fibril growth kinetics in an isoform-specific manner. These findings will be informative and useful to future research on tau PTMs, even though currently deeper conclusions about these data are somewhat difficult to make due to the lack of structural information.

The only remaining problem is that the cross-seeding data stand out as superficial. Between two tau isoforms, there can be a total of 4 cross-seeding combinations: 1) unmodified 3R tau seeding unmodified 4R tau; 2) acetyl-3R tau seeding unmodified 4R tau; 3) unmodified 3R tau seeding acetyl-4R tau; 4) acetyl-3R tau seeding acetyl-4R tau. The authors studied samples #2 and sample #4, and found that seeding worked for #2 but not for #4. The significance of this difference is unclear, due to the lack of structural information. Why is unmodified 4R tau malleable to templating by acetyl-3R tau (#2)? Wouldn't the presence of R2 domain in 4R tau interfere with seeding by acetyl-3R tau? What is special about the acetyl-3R tau structure? Would unmodified 3R tau be necessarily worse than acetyl-3R tau in seeding unmodified 4R tau? To prove that the resistance of acetyl-4R tau to seeding by acetyl-3R tau is important for disease, one would need to prove that neither unmodified 4R tau nor unmodified 3R tau is relevant to disease. This part of the

study opens more question than it answers, and could form a study in itself. I do not think this seeding result is essential to the main conclusion of the paper (which are that acetylation changes aggregation kinetics in an isoform-specific way, and that there are hotspots for Lys acetylation), so I recommend that the authors remove this section and focus on the many other clearer findings.

Reviewer #1

The manuscript of Chakraborty et al has been revised substantially according to my comments. The new experiments that have been done highlight a role of auto-acetylation by significantly modulating the acetylation pattern of acetyltransferases CBP/p300. Notably, the major difference due to auto-acetylation resides in the acetylation levels of lysines in the R2 repeat which is probably the main reason why acetylated tau3R and tau4R behave differently in the aggregation reactions.

Reply: We thank the reviewer for the careful evaluation of the manuscript

In whole, all lysine residues appear to be acetylated (without any exception) at relatively low levels (less than 25%) except those of the R2/R3 repeats around the cysteine residues that display high acetylation levels. Moreover, the autoacetylation levels are comparable if not higher than enzymatic levels in most places which indicates the auto-acetylation is the main driver of acetylation here (see Fig 2d). My opinion is that the determination of the acetylation levels on NMR spectra by comparison of relative intensities between acetylated vs. non-acetylated samples is inappropriate as it clearly appears by looking directly on the spectra of Fig 2b,c that there are not so many acetylation sites. The intensity changes btw. Ac vs. non-Ac tau could be explained by conformational changes and/or differential water exchange of amide protons. This should be clarified by throughout investigation and above all, assigning the acetylation sites (or compare with published data showing the assignment of acetylated Tau spectrum) should provide a direct quantification of acetylation levels.

Reply: We thank the reviewer for this comment. As stated by the reviewer, this comment was probably triggered by the spectra shown in Fig. 2bc. However, abundant acetylation becomes apparent when the spectra are plotted such that also weaker cross-peaks become visible:

In addition, we now added mass spectrometry data to the manuscript (Supplementary Fig. 3a and Supplementary Table 1) demonstrating that most if not all lysine residues of 4R tau were acetylated, in agreement with our finding from the NMR analysis.

Regarding the determination of the acetylation level, we agree that conformational changes or differential water exchange may affect the intensities thereby affecting the determination of the acetylation levels. However, such effects may also affect the quantification of the acetylation levels from the intensity of the acetylated peaks as those resonances will also be influenced by potential conformational changes or differential water exchange. In addition, it would require the assignment of many acetylated lysine resonances, which is highly challenging as almost all the lysine residues are acetylated and the presence of acetylation-induced peak splitting and signal overlap. In contrast, the acetylation level can still be determined within a reasonable error limit if we make use of the intensity of the unacetylated residues (this is not ideal for the residues that are very close in the protein sequence like K224/K225, K280/K281, K369/K370; we already acknowledge this limitation on page 5).

For the revision, we also recorded HSQC spectra of the ¹⁵N-labeled unmodified and acetylated 4R tau and calculated the residue-specific intensity ratio by dividing the intensity of the cross-peaks observed in the acetylated sample by the unmodified sample (Supplementary Fig. 3b). The intensity ratio plot shows a local intensity decrease at the position of lysine residue, but the region without the lysine residues does not show any intensity decrease. This suggests that any potential global conformational change of the protein upon acetylation does not strongly affect the analysis.

Another point concerns the acetylation mimics. The lysine-to-glutamine mutation is generally used to mimic the acetylation state (in part by neutralization of the positive charge), e. g. as shown by Min et al., *Nature Medicine* (2015) 21: 1154–1162 or Ajit et al, *JBC* (2019) 294(45):16698-16711. However, the K-to-Q mutants (5 out of 6 mutants) aggregate faster than wild-type 4R tau while acetylation was shown to delay 4R tau aggregation as also illustrated by pseudo-acetylation mimics. This discrepancy should be clearly explained.

Reply: Thanks for the comment. To clarify this issue, we state on page 12: “*K-to-Q mutation is widely used to mimic acetylation of proteins in cell and animal studies*^{29,30}. The mutation captures the acetylation-associated removal of the positive charge of lysine, but fails to represent the size of the acetylated lysine side chain (Supplementary Fig. 7). We therefore created four single-site acetylation mimics (K280Ac, K294Ac, K298Ac, K311Ac) of 4R tau by mutating individual lysine residues in R2 to cysteine, followed by converting the cysteine residue to dehydroalanine, which enables the access of N-acetylcysteamine (Supplementary Fig. 7)³¹.”

Based on the data presented here, I cannot definitely see the benefits of ssNMR on the structural description of acetylated tau fibrils. Given the limited information provided by ssNMR experiments, the “effect of acetylation on the structure of 3R tau fibrils” as indicated in the subtitle is largely speculative. The pronase digestion profile and MS analyses are more convincing. A re-organization of the manuscript on this specific point must be done putting all these data together.

Reply: Thanks for this suggestion. In the revised version of the manuscript, we reorganized Fig. 5 by combining the MS and solid-state NMR data.

Reviewer #2

The authors have addressed most of my original comments. Overall, the manuscript reports many interesting observations about how in vitro acetylation levels vary across the Lys residues, and how acetylation changes fibril growth kinetics in an isoform-specific manner. These findings will be informative and useful to future research on tau PTMs, even though currently deeper conclusions about these data are somewhat difficult to make due to the lack of structural information.

Reply: We thank the reviewer for the encouraging comments.

The only remaining problem is that the cross-seeding data stand out as superficial. Between two tau isoforms, there can be a total of 4 cross-seeding combinations: 1) unmodified 3R tau seeding unmodified 4R tau; 2) acetyl-3R tau seeding unmodified 4R tau; 3) unmodified 3R tau seeding acetyl-4R tau; 4) acetyl-3R tau seeding acetyl-4R tau. The authors studied samples #2 and sample #4, and found that seeding worked for #2 but not for #4. The significance of this difference is unclear, due to the lack of structural information. Why is unmodified 4R tau malleable to templating by acetyl-3R tau (#2)? Wouldn't the presence of R2 domain in 4R tau interfere with seeding by acetyl-3R tau? What is special about the acetyl-3R tau structure? Would unmodified 3R tau be necessarily worse than acetyl-3R tau in seeding unmodified 4R tau? To prove that the resistance of acetyl-4R tau to seeding by acetyl-3R tau is important for disease, one would need to prove that neither unmodified 4R tau nor unmodified 3R tau is relevant to disease. This part of the study opens more question than it answers, and could form a study in itself. I do not think this seeding result is essential to the main conclusion of the paper (which are that acetylation changes aggregation kinetics in an isoform-specific way, and that there are hotspots for Lys acetylation), so I recommend that the authors remove this section and focus on the many other clearer findings.

Reply: We thank the reviewer for this suggestion. Accordingly, we removed the figures and discussions related to the cross-seeding of tau.

Reviewers' Comments:

Reviewer #1:

Remarks to the Author:

The manuscript of Chakraborty et al has been revised according to my comments. New data has been added in Supplementary Information.

I have no additional comment.